# The Snowflake Hypothesis: Training Deep GNN with One Node One Receptive field

## Abstract

Despite Graph Neural Networks (GNNs) demonstrating considerable promise in graph representation learning tasks, GNNs predominantly face significant issues with overfitting and over-smoothing as they go deeper as models of computer vision (CV) realm. Given that the potency of numerous CV and language models is attributable to that support reliably training very deep architectures, we conduct a systematic study of deeper GNN research trajectories. Our findings indicate that the current success of deep GNNs primarily stems from (I) the adoption of innovations from CNNs, such as residual/skip connections, or (II) the tailor-made aggregation algorithms like DropEdge. However, these algorithms often lack intrinsic interpretability and indiscriminately treat all nodes within a given layer in a similar manner, thereby failing to capture the nuanced differences among various nodes. In this paper, we introduce *the Snowflake Hypothesis* – a novel paradigm underpinning the concept of "one node, one receptive field". The hypothesis draws inspiration from the unique and individualistic patterns of each snowflake, proposing a corresponding uniqueness in the receptive fields of nodes in the GNNs.

We employ the simplest gradient and node-level cosine distance as guiding principles to regulate the aggregation depth for each node, and conduct comprehensive experiments including: (1) different training scheme; (2) various shallow and deep GNN backbones, especially on deep frameworks such as JKNet, ResGCN, PairNorm *etc.* (3) various numbers of layers (8, 16, 32, 64) on multiple benchmarks (six graphs including dense graphs with millions of nodes); (4) compare with different aggregation strategies. The observational results demonstrate that our framework can serve as a universal operator for a range of tasks, and it displays tremendous potential on deep GNNs. It can be applied to various GNN frameworks, enhancing its effectiveness when operating in-depth, and guiding the selection of the optimal network depth in an explainable and generalizable way.

## 1 Introduction

Graph Neural Networks (GNNs) (Kipf & Welling, 2017; Hamilton et al., 2017) have emerged as the leading models for various graph representation learning tasks, including node classification (Velickovic et al., 2017; Abu-El-Haija et al., 2020), link prediction (Zhang & Chen, 2018; 2019), and graph classification (Ying et al., 2018; Gao & Ji, 2019). The prominent performance of GNNs GNNs primarily derives from their message passing mechanism (Wu et al., 2020). This mechanism, operating iteratively, adeptly garners informative representations by aggregating knowledge from neighboring nodes within the graph topology (Wu et al., 2020).

Though promising, overfitting (Rong et al., 2019), over-smoothing (Li et al., 2018; Chen et al., 2020a) and vanishing gradients (Li et al., 2019; Zhao & Akoglu, 2019) are three long-standing problems in the GNN area, especially when GNNs go deeper as Convolutional Neural Networks (CNNs) (He et al., 2016). Consequently, when training an over-parameterized GNN on a small graph or utilizing a deep GNN for graph modeling, we often end up with collapsed weights or indistinguishable node representations (Chen et al., 2020a). Therefore, training 2-to-4-layer GNNs is not a foreign phenomenon in the graph realm and most state-of-the-art GNNs are no deeper than 4 layers (Sun et al., 2019). In contast, the brilliant achievements on many computer vision tasks can be primarily attributed to the consistent and effective training of deep networks (He et al., 2016; Huang et al., 2017). Graph representation learning eagerly calls for the utilization of deeper GNNs, particularly when dealing with large-scale graphs characterized by dense connections.

Remarkably, a number of recent endeavors have demonstrated the feasibility of training GNNs with progressively increasing depth. We can summarize the existing approaches into two categories: The first category involves prudently inheriting innovations from CNNs, such as residual/dense connections (Li et al., 2019; Sun et al., 2019; Xu et al., 2018; Li et al., 2021; Xu et al., 2018; Chen et al., 2020b; Xu et al., 2021), which have proven to be universally applicable and practical. For instance, JKNet (Xu et al., 2018) adopts skip connections to fuse the output of each layer to maintain the discrepancies among different nodes. GCNII (Chen et al., 2020b) and ResGCN (Li et al., 2019) employ residual connections to carry the information from the previous layer to avoid the aforementioned issues. Another category is to combine various deep aggregation strategies with shallow neural networks (Wu et al., 2019; Chien et al., 2020; Liu et al., 2020; Zou et al., 2019; Rong et al., 2019; Gasteiger et al., 2019). For example, GDC (Gasteiger et al., 2019) generalizes Personalized PageRank into a graph diffusion process. DropEdge (Rong et al., 2019) resorts to a random edge-dropping strategy to implicitly increase graph diversity and reduce message passing.

While CNN inheritances such as residual/skip connections can partially alleviate the over-smoothing problem, these modifications fail to effectively explore the relationship between aggregation strategies and network depth. Incorporating residuals into layers with suboptimal outputs may inadvertently propagate detrimental information to subsequent aggregation layers. Within the second category, the majority of existing deep aggregation strategies attempt to sample a subset of neighboring nodes around the central node to implicitly enhance data diversity and prevent over-smoothing. Unfortunately, the cumbersome and particular designs make GNN models neither simple nor practical, lacking the ability to scale on other training strategies and specific datasets.

In this paper, we hypothesize that each node within a graph should possess its unique receptive field. This can be actualized through the process of element-level adjacency matrix pruning. Such a procedure enables an "early stopping" feature in node aggregation in terms of depth, which not only amplifies interpretability but also aids in mitigating the over-smoothing issue. Drawing from the extensive experimental observations, we first introduce *the Snowflake Hypothesis* (SnoH).

**The Snowflake Hypothesis (SnoH).** When learning on an abstracted graph $\mathcal{G} = (\mathcal{V}, \mathcal{E})$ with a vertice set $\mathcal{V}$ and an edge set $\mathcal{E}$ from the natural world using GNN models[1], by utilizing a masked adjacency matrix to prune the graph structure, we can uncover the distinctive receptive fields that each node ought to aggregate, which is akin to the uniqueness observed in each snowflake. This enables us to overcome the over-smoothing problem when training deeper GNNs. We summarize the philosophy of SnoH in Fig 1 for ease of understanding.

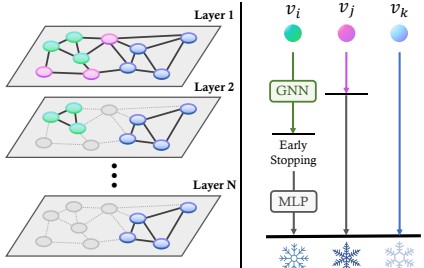

**Figure 1:** The philosophy of SnoH.

We utilize the simplest gradient (denoted as Version 1) and node-level cosine distance (denoted as Version 2) as the guiding principles to control the aggregation depth for each node, and we adopt extensive experiments on node classification with various settings, including 1) **Different training algorithms** such as a pre-training scheme (Liu et al., 2023), iterative pruning (Frankle & Carbin, 2018; Chen et al., 2021), and re-initialization. We found that compared to popular pruning algorithms, our approach not only allowed us to leverage the benefits of pruning but also facilitated better network convergence. 2) **Integration with various deep GNN models**, such as ResGCN (Li et al., 2021), JKNet (Xu et al., 2018), and GCNII (Chen et al., 2020b). This integration empowers us to more effectively aid the model in enhancing its performance and facilitating deeper architectures. 3) **In comparison with DropEdge or graph lottery ticket methods** (Chen et al., 2020c), we observe that our algorithm possesses better interpretability and generalizability. Our algorithm can also be conceptualized as a data augmentation approach and a message passing reducer (Rong et al., 2019), providing a new paradigm to benefit many graph pruning applications.

**Identifying unique snowflake.** We identify a unique snowflake by employing layer-wise adjacency matrix pruning and node-level cosine distance discrimination. Taking a 2-layer vanilla GCN (Velickovic et al., 2017) as example, we assume that the trainable parameters $\Theta = \{\Theta^{(0)}, \Theta^{(1)}\}$ for node

---

[1]GNNs aim to learn a representation vector of a node or an entire graph based on the adjacency matrix $A \in \mathbb{R}^{|\mathcal{V}| \times |\mathcal{V}|}$ and node features $X \in \mathbb{R}^{|\mathcal{V}| \times F}$.

classification as:

$$\mathcal{Z} = \text{Softmax}\left(\hat{A}\sigma\left(\hat{A}X\Theta^{(0)}\right)\Theta^{(1)}\right), \quad \text{loss function: } \mathcal{L}\left(\mathcal{G}, \Theta\right) = -\sum_{v_i \in \mathcal{V}_l} y_i \log\left(z_i\right)$$

where $\mathcal{Z}$ is the model predictions, $\sigma\left(\cdot\right)$ denotes an activation function, $\hat{A} = \tilde{S}^{-\frac{1}{2}}\left(A + I\right)\tilde{S}^{-\frac{1}{2}}$ is the normalized adjacency matrix with self-loops and $\tilde{S}$ is the degree matrix of $A + I$. We minimize the cross-entropy loss $\mathcal{L}\left(\mathcal{G}, \Theta\right)$ over all labelled nodes $\mathcal{V}_l \subset \mathcal{V}$, where $y_i$ and $z_i$ represents the label and prediction of node $v_i$, respectively. We present the first versions (v1) of SnoH as follows:

1. Randomly initialize a GNN denoted as $f\left(\mathcal{G}, \Theta_0\right)$ for learning on the graph $\mathcal{G}$.

2. Train the GNN (with totally $D$ layer) for $k$ iterations; compute the absolute gradient of each element in the outermost adjacency matrix (denote as $A_{(D)}$); remove the elements with the top-$p\%$ smallest gradients in the adjacency matrix.

3. Repeat Step 2 by computing $A_{(D-1)}$ and assigning the index of zero elements in $A_{(D-1)}$ to $A_{(D)}$, thereby setting corresponding positions of the next layer's adjacency matrix to zero.

4. Repeat Step 2 & 3 iteratively; remove the corresponding elements from the adjacency matrix and assign their zero-element positions to all deeper layers.

By utilizing the above gradient guidance which can indicate potentially promising elements (Le et al., 2020; Blalock et al., 2020), we can realize the Snowflake Hypothesis, enabling each node to have its unique aggregation depth and receptive field size. For ease of understanding, we showcase a more comprehensive training implication in Appendix A. However, the task of calculating gradients for the adjacency matrix presents a substantial challenge, especially for larger graphs with millions of nodes such as Ogbn-product that possesses 61,859,140 edges (Hu et al., 2020). Blindly calculating the gradient for each edge is both difficult and unwise. The sheer volume of parameters involved poses a formidable obstacle when attempting to integrate this algorithm into deep GNNs. To tackle this issue, we further present the second version, *i.e.*, SnoHv2 in which we make a simple modification to focus on node representations:

1. Randomly initialize a GNN denoted as $f\left(\mathcal{G}, \Theta_0\right)$ for learning on the graph $\mathcal{G}$.

2. Train the $D$-layer GNN for $k$ iterations; calculate the cosine distance $D(Z^{(l)}, \mathcal{T}(Z^{(l)})) = 1 - \frac{Z^{(l)} \cdot \mathcal{T}(Z^{(l)})}{||Z^{(l)}||_2 \cdot ||\mathcal{T}(Z^{(l)})||_2}$ between the representation of each node in the GNN and its aggregated representation of the surrounding nodes at each layer; here $Z^{(l)}$ denotes the representation of the $i$-th node at the $l$-th layer before aggregation by the adjacency matrix, while $\mathcal{T}$ symbolizes the representation after aggregation. In the context of GCN, $Z^{(l)}$ is defined as $H^{(l)} \cdot W^{(l)}$, and $\mathcal{T}(Z^{(l)})$ is represented as $A^{(l)}H^{(l)}W^{(l)}$, where $H^{(l)}$ is the feature embedding at the $l$-th layer.

3. Compute the nodes whose cosine distance (Le et al., 2020; Blalock et al., 2020) is below $p\%$ of the distance at the initial layer, and remove the element for node aggregation corresponding to these nodes. As an example, for the $i$-th node, this operation is equivalent to pruning all elements in the $i$-th row of the adjacency matrix (excluding self-loops that are not pruned).

4. Once the cosine distance of the $i$-th node at the $r$-th layer falls below $p\%$ of the distance at the initial layer, all elements within the $i$-th row of the adjacency matrices in all subsequent layers will undergo pruning.

The rationale behind SnoHv2 is rather straightforward: *As the depth of GNNs increases, the issue of over-smoothing becomes more severe. Representations of neighboring nodes tend to converge, which in turn leads to the network losing its discriminative capacity. Implementing early stopping in terms of depth can aid in restoring the expressiveness of the nodes.*

## 2 IMPLEMENTATIONS & CONTRIBUTIONS

In our paper, we aim to examine the effectiveness of SnoH over various training schemes, backbones, and datasets. We also integrate our approach with state-of-the-art (SOTA) deep GNNs and compare it with similar popular algorithms to further demonstate its scalability and generality.

**Training schemes.** We select three training methods to explore the performance of our algorithm and the benefits of combining our algorithm with mainstream training approaches: (1) **SnoHv1/v2(O)**: we adopt the original *hierarchical one-shot adjacency pruning* approach. (2) **SnoHv1/v2(IP)**: As our work can be regarded as a graph pruning method, we have opted for the

widely recognized iterative pruning (IP) strategy (Frankle & Carbin, 2018) within the framework of unified graph sparsification (UGS) (Chen et al., 2020c). (3) **SnoHv1/v2(ReI)**: Due to the challenge of determining whether the model has converged during pruning, after pruning an adjacency matrix, we fix it and *reinitialize* the GNN for the next training phase. We place the details in Appendix B.

**Datasets & Backbones.** In this paper, we utilize six graph benchmarks to evaluate the performance of our hypothesis. Specifically, we select three widely-used small graphs, namely Core, citeseer, and PubMed (Kipf & Welling, 2017), for node classification. Additionally, to assess the scalability of our proposal, we incorporate three large-scale graphs known as Ogbn-arxiv, Ogbn-proteins and Ogbn-products (Hu et al., 2020). For all selected datasets, we compare our framework with different baseline settings under the same network configurations. We adopt GCN (Kipf & Welling, 2017), GIN (Xu et al., 2019) and GAT (Veličković et al., 2017) as shallow GNNs backbones. Further, we take deep ResGCNs (Li et al., 2020), JkNet (Xu et al., 2018) and PairNorm (Zhao & Akoglu, 2019) as deep backbones. To evaluate our framework on the graph classification task, DropEdge (Rong et al., 2019) and UGS (Chen et al., 2020c) are leveraged as the comparison algorithms. More details about experimental settings can be found in Appendix C.

**Contributions.** We summarize our contributions as three folds:

- We propose a node "early stopping" technique based on edge pruning to help better combat the issue of over-smoothing and overfitting. Based on extensive observational results, we put forth "The Snowflake Hypothesis – one node one receptive field", which is inspired by the notion that each snowflake is unique and possesses its own pattern. Likewise, each node in GNNs should have its own receptive field, reflecting its unique characteristics.

- Our algorithm inherently possesses explainability and, while inheriting the advantages of the pruning algorithms (accelerating inference time and reducing storage overhead), it can also benefit the current graph pruning algorithms. More importantly, our algorithm is simple and convenient. Compared to developing complex aggregation strategies, our framework does not introduce any additional information (*e.g.*, learnable parameters), which can be easily scaled up to deep GNNs.

- We conduct extensive experiments, i.e., spanning a series of training algorithms, integration with various backbone architectures, and comparisons with DropEdge/UGS frameworks, across multiple graph benchmarks. The results show that SnoHv1/v2 consistently delivers standout performance, even in scenarios where the adjacency matrix is notably sparse. This underscores our initial hypothesis – certain nodes necessitate early termination in their depth progression.

**Prior work.** Our work contributes to the domain of graph pruning algorithms, aligning with the research trajectories of graph sampling (Chen et al., 2018; Eden et al., 2018; Chen et al., 2021) and graph pooling (Ranjan et al., 2020; Zhang et al., 2021; Ying et al., 2018). In particular, our algorithm shares significant parallels with the prevalent graph lottery ticket pruning algorithm (Chen et al., 2020c), striving to replicate the performance of an original, unpruned graph by means of iterative pruning. Moreover, we aim to address the over-smoothing and overfitting problems that may surface during the training of deep GNNs (Li et al., 2018; Chen et al., 2020a). Current methodologies in training deep GNNs principally concentrate on two areas: (1) incorporating components such as residual/skip connections from the architecture of CNNs (Li et al., 2019; Sun et al., 2019; Xu et al., 2018) , and (2) crafting a diverse array of aggregation strategies (Wu et al., 2019; Chien et al., 2020; Liu et al., 2020; Zou et al., 2019). These focal areas also form the bedrock of our proposed research framework. We refer detailed discussions in Appendix D.

## 3 IDENTIFYING THE UNIQUE SNOWFLAKES IN SMALL GRAPHS

In this section, we meticulously conduct a multitude of experiments to validate our hypothesis on several small graphs, namely Cora, citeseer, and PubMed. We choose **SnoHv1/v2(O)** as the benchmark training scheme. The experimental settings are placed in Appendix E.

As depicted in Tab 1, under conditions of deep GNNs (especially for high graph sparsity at deep layers), SnoHv1 can achieve results approximating those of the original baseline. This observation indicates that implementing early stopping for certain nodes in terms of depth does not compromise the overall performance of the model. Upon transitioning to the more robust SnoHv2 version (Fig 2), we notice a performance enhancement in our model. This further suggests that early stopping in depth may help overcome the over-smoothing phenomenon. As frameworks like ResGCN and JKNet are specifically designed for deep GNNs, we have not presented results for shallow layers. Here, we independently document the results of SnoHv2 for shallow layers. In the case of a 2-layer

**Table 1:** Performance comparisons on 8, 16, 32 layer settings using SnoHv1/v2 across three small graphs. All experimental results are the average of **five runs** and the red font indicates the optimal value in a set of results.

| Backbone | 8 layers | | 16 layers | | 32 layers | | 64 layers | |
|---|---|---|---|---|---|---|---|---|
| | Original | SnoHv1/v2 | Original | SnoHv1/v2 | Original | SnoHv1/v2 | Original | SnoHv1/v2 |
| *Train scheme: SnoHv1/v2(O), Dataset: Cora, 2-layer performance: GCN without BN = 85.37* | | | | | | | | |
| GCN | 85.11 | 85.17/85.68 | 83.75 | 83.87/84.19 | 80.33 | 81.10/83.09 | 66.11 | 68.45 /72.88 |
| ResGCN | 85.31 | 85.37/86.11 | 85.75 | 85.99/86.52 | 86.27 | 86.33/86.64 | 85.21 | 85.24/85.90 |
| JKNet | 86.33 | 87.01/86.53 | 86.28 | 86.17/87.08 | 87.20 | 87.31/88.86 | 84.84 | 85.19/85.96 |
| PairNorm | 83.66 | 82.11/85.90 | 80.29 | 80.44/83.25 | 78.66 | 79.34/83.16 | 74.12 | 74.19/78.60 |
| *Train scheme: SnoHv1/v2(O), Dataset: Citeseer, 2-layer performance: GCN without BN = 72.44* | | | | | | | | |
| GCN | 72.39 | 72.41/73.24 | 71.28 | 72.10/72.33 | 68.99 | 69.21/69.89 | 44.37 | 45.12/46.65 |
| ResGCN | 72.11 | 72.07/72.23 | 72.40 | 71.91/71.91 | 72.43 | 72.44/73.53 | 71.65 | 72.10/72.94 |
| JKNet | 71.77 | 71.50/71.47 | 70.72 | 70.60/71.47 | 70.12 | 70.01/72.67 | 69.92 | 70.09/71.55 |
| PairNorm | 72.88 | 72.34/73.84 | 73.91 | 73.95/74.58 | 73.36 | 73.05/73.92 | 70.88 | 70.85/72.99 |
| *Train scheme: SnoHv1/v2(O), Dataset: PubMed, 2-layer performance: GCN without BN = 86.50* | | | | | | | | |
| GCN | 86.41 | 86.50/86.56 | 84.77 | 84.74/85.79 | 83.76 | 83.77/84.06 | 77.29 | 78.15/78.99 |
| ResGCN | 87.45 | 87.50/87.84 | 87.73 | 87.47/88.33 | 87.66 | 87.33/88.49 | 87.01 | 86.03/88.11 |
| JKNet | 88.20 | 88.31/88.51 | 87.32 | 87.62/87.95 | 88.81 | 88.75/88.99 | 87.25 | 86.98/87.93 |
| PairNorm | 87.63 | 87.50/88.68 | 87.92 | 87.74/88.60 | 87.07 | 87.24/87.69 | 85.41 | 85.48/87.06 |

**Figure 2:** Performance comparisons on 32/64-layer settings using SnoHv2 across three small graphs.

GCN on Cora, we observe a score of 86.08% (baseline 85.37%), on Citeseer, it's 73.81% (baseline 72.44%), and on PubMed, it's 88.54% (baseline 86.50%). Tab 1 shows that even in shallow GCN, implementing "early stopping" for certain nodes in depth could enhance performance (0.29%↑ on Cora and 0.23%↑ on Citeseer). With regard to PubMed, we argue that due to the relative largeness, even after two layers of aggregation, better representations may not have been learned. All nodes may require a deeper receptive field, which aligns with the phenomenon observed in the table where extending the depth to between 8-32 layers leads to a performance boost after pruning.

We have another interesting observation: when the depth of the GCN reaches 32/64 layers, SnoHv2 shows a stronger performance boost. Under the experimental setup of a 64-layer GCN + SnoHv2, improvements of 6.77%/1.66%/1.12% are achieved on Cora/Citeseer/PubMed, respectively. These astonishing results clearly verify the effectiveness of our algorithm. In Tab 7 of Appendix E, we present the sparsity under different datasets. As the network goes deeper, both node sparsity[2] and edge sparsity are decreasing. At a lower level with high sparsity (around 17%∼32%), certain nodes and edges were pruned, resulting in an improvement in model accuracy. This validates the contribution of reducing the receptive field in shallow layers. In Tab 8, the 64-layer GNN on Cora can reach 6.57% node sparsity and 15.26% edge sparsity in deep layers, which further indicates that many nodes in deep networks do not necessitate such an extensive receptive field.

---

[2] Node sparsity denote the ratio of nodes that do not aggregate information from their neighborhood (*i.e.*, its degree is zero at this layer) to the total number of nodes in the graph.

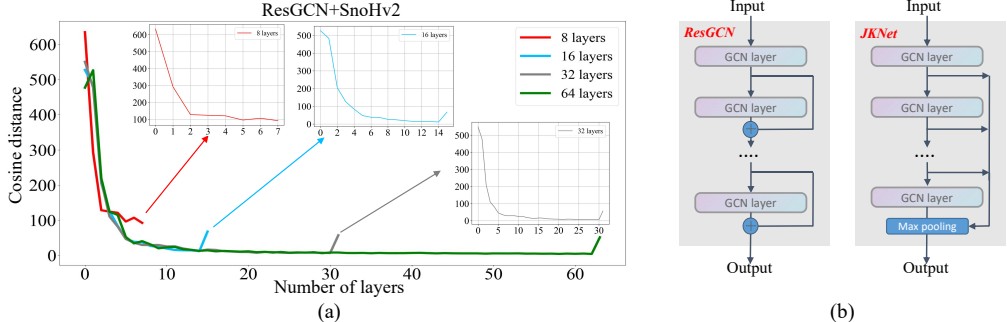

**Figure 3:** (a) Cosine distances of ResGCN+SnoHv2 on Cora vs. #layers. (b) Pipeline of ResGCN and JKNet.

**How does SnoH help deep GNNs?** Our investigations have revealed that the integration of our framework yields discernible benefits for deep GNNs such as ResGCN, JKNet, and PairNorm, which are summarized as follows. **Obs 1.** In deep architectures (e.g., with 32 or 64 layers), SnoH brings more significant improvements to ResGCN than in shallow architectures. For instance, ResGCN achieves an accuracy of 85.21%, 71.65%, and 87.01% on Cora, Citeseer, and PubMed at 64 layers. When combined with SnoHv2, its performance is enhanced to 85.90%, 72.94%, and 88.11%, as shown in Fig 10 of Appendix E. **Obs 2.** Taking ResGCN and JKNet as examples (Fig 3 (b)), the sparsity of each layer can surprisingly help us determine the specific depth that the GCN architecture should retain. For instance, in the case of JKNet+Cora, the edge sparsity decays to zero after the 13th layer. This indirectly indicates that the aggregation output of the adjacency matrix after the 16th layer no longer contributes to the model, and similar phenomena occur under different combination designs, such as ResGCN+Citeseer with fewer than 26 layers. **Obs 3.** Fig 3(a) depicts the cosine distance for each layer of ResGCN+SnoHv2 settings with various depths of GNN on Cora. We witness a gradual decrease in cosine distance with increasing depth, furnishing additional substantiation for the presence of the over-smoothing phenomenon. This leads to the convergence of node representations as the network deepens, consequently impeding predictive performance. To surmount the over-smoothing challenge and augment interpretability, we implement a pruning strategy to curtail the depth of node expansion within the network. Additional results are presented in Fig 11&12 of Appendix E.

**Compare with pruning algorithm.** Our algorithm can be essentially viewed as an adjacency matrix pruning algorithm. We thus choose a SOTA graph pruning algorithm (*i.e.*, UGS (Chen et al., 2020c)) and a universal method (*i.e.*, random pruning) for comparison. To keep consistent settings, we removed the part of UGS that targets weight pruning; we control the iterative pruning rates at 5%, 10%, and 20% and prune 5 times. In order to make a better comparison, we record the pruning rates when discovering tickets in the lottery ticket scenario, and the pruning rates of SnoHv2 when it gets the best performance for comparison. We showcase the comprehensive results in Tab 12 (Appendix E) and summarize our findings: **Obs 1.** Our model can clearly surpass random pruning and UGS, even under higher sparsity levels. This further validates our performance in deeper GNNs, providing assurance for our algorithm's scalability on large datasets. **Obs 2.** In contrast to UGS, our method exhibits superior interpretability. UGS maintains a consistent graph sparsity level for each layer, potentially resulting in the elimination of numerous nodes during early training stages. This is unreasonable as early nodes should aggregate essential information, ensuring they can learn better representations. Our experiments further confirm that the receptive field should gradually increase.

We also migrate potential training strategies in pruning to SnoHv1 and display the results in Appendix E. As can be easily seen, different training strategies do not significantly improve SnoHv1's results. However, in practice, iterative pruning and re-initialization strategies can bring about severe efficiency problems, *e.g.*, even $D\times$ on the training burden of re-initialization. Hence, we adopt a

**Table 2:** Comparison performances of SnoHv2 with UGS and random pruning (RP). Here IPR denotes iterative pruning rate and we set number of layers as 8. We use GCN backbone and set early stopping threshold of cosine distance as $\rho$ (Detailed descriptions in Appendix E).

| Dataset | RP | UGS(IPR=5%) | UGS(IPR=10%) | UGS(IPR=20%) | SnoHv2 | GCN |
|---------|------|-------------|--------------|--------------|--------|-------|
| Cora (L=8) | 69.60 | 73.64 | 66.01 | 53.29 | 85.68 | 85.11 |
| Citeseer (L=8) | 45.50 | 65.80 | 51.50 | 43.10 | 73.24 | 72.39 |
| PubMed (L=8) | 77.82 | 84.33 | 80.91 | 71.05 | 86.56 | 86.41 |

one-shot pruning strategy as the preferred strategy for SnoHv1. Unless specified otherwise, the performance we present is that of SnoHv1(O). In terms of SnoHv2 which fundamentally determines the early stopping depth of each node by similarity, different training strategies will not have much significance in this case.

**Comparision with DropEdge.** Another related approach is edge dropping, *e.g.*, DropEdge (Rong et al., 2019) shares similarities with SnoH. Although DropEdge can improve performance through implicit data augmentation, it lacks interpretability in its aggregation strategy. In fact, it should not continue aggregation after halting the information aggregation for a certain node at a higher layer. It is worth noting that since DropEdge involves temporarily increasing data samples during training, it can be easily combined with SnoHv2. We present the comparison results in Tab 3.

**Generalizability on different backbones.** To validate the generalization capability of our algorithm across different backbones, we further selected popular GNN frameworks GIN (Xu et al., 2019) and GAT (Veličković et al., 2017) as the backbones for the generalization evaluation. We control the parameter $\rho$ in {0.2, 0.1, 0.05} under experiments with network depths of 8, 16 and 32 layers, respectively. As depicted in Fig 4, our hypothesis remains viable when applied to GIN and GAT. When combined with SnoHv2, these backbones still demonstrate improved performance in deeper layers. Specifically, we achieve a gain of 2.46% and 0.73% gains on a 16-layer GIN and a 64-layer GAT, respectively. These results further support the generalization capability of our algorithm.

| | Total layers | Graphs | | |
|---|---|---|---|---|
| | | Cora | Citeseer | PubMed |
| DropEdge | 8 | 86.98 | 74.57 | 86.91 |
| | 16 | 84.01 | 73.17 | 86.35 |
| | 32 | 80.81 | 71.77 | 82.23 |
| DropEdge + SnoHv2 | 8 | 81.70 | 72.97 | 87.25 ↑ |
| | 16 | 80.32 | 72.12 | 86.99 ↑ |
| | 32 | 76.64 | 65.56 | 87.19 ↑ |

**Table 3:** Comparison between DropEdge and DropEdge+SnoHv2.

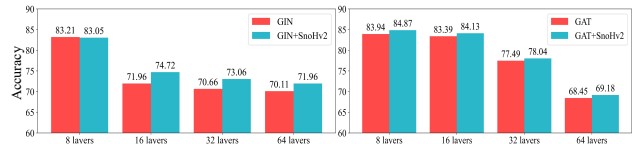

**Figure 4:** Experiment results on different graph backbones (*i.e.*, GIN, GAT) over the Cora dataset. Additional results on Citeseer and PubMed can be found in Appendix F.

## 4 SNOHV1/V2 ON LARGE-SCALE GRAPHS

In this section, we thoroughly assess our hypothesis on large graphs, employing three benchmark datasets: Ogbn-Arxiv, Ogbn-Proteins, and Ogbn-Product. The dataset partition adheres to the guidelines provided by (Hu et al., 2020). For Ogbn-ArXiv, our training data comprises papers published until 2017, validation data encompasses papers published in 2018, and testing data includes papers published since 2019. Regarding Ogbn-Proteins, we partition proteins into training, validation, and test sets based on their species. For the Product dataset, we adopt sales ranking as the criterion to divide nodes into training, validation, and test sets. Specifically, we assign the top 8% of products to the training set, the next top 2% to the validation set, and the remainder to the test set.

### 4.1 SNOHV1/V2 ON CITATION NETWORK

We first investigate the Snowflake Hypothesis on Ogbn-Arxiv, a dataset that serves as a representative example of real-world graph scenarios. Specifically, we consider GCN, ResGCN and ResGCN+ (Li et al., 2020) as backbones for evaluation. Among them, we prune the adjacency matrix each layer separately guided by the cosine distance. Due to the superior ability of ResGCN and ResGCN+ to mitigate over-smoothing compared to GCN, we adopt larger values of $\rho$ on ResGCN and ResGCN+ networks. Specifically, for ResGCN and ResGCN+, we use threshold values of 0.1, 0.07, and 0.02 at layers 16, 32, and 64, respectively. For GCN, we use thresholds of 0.08, 0.05 for 16 and 32 layers. The experimental results are shown in Fig 5, from which we can conclude: **Obs 1.** SnoHv2, when combined with three types of backbones, can achieve the same or even better performance under relatively high sparsity conditions as compared to the original network. **Obs 2.** SnoHv1 can achieve a slightly higher improvement (Tab 4) compared to SnoHv2 on the citation network. Through heterogeneity analysis (Pei et al., 2020), the citation network possesses relatively more severe heterogenous networks, and the comparison at different levels might be of relatively low significance for this type of graph; we should rather avoid the aggregation of heterogenous information from the initial layers. We have placed specific analysis and conjecture in Appendix G.

Similarly, we follow UGS with a 28-layer ResGCN+Arxiv as the benchmark setting, and remove the weight pruning part. We iteratively prune at a rate of 0.05 for 20 times, observing the sparsity

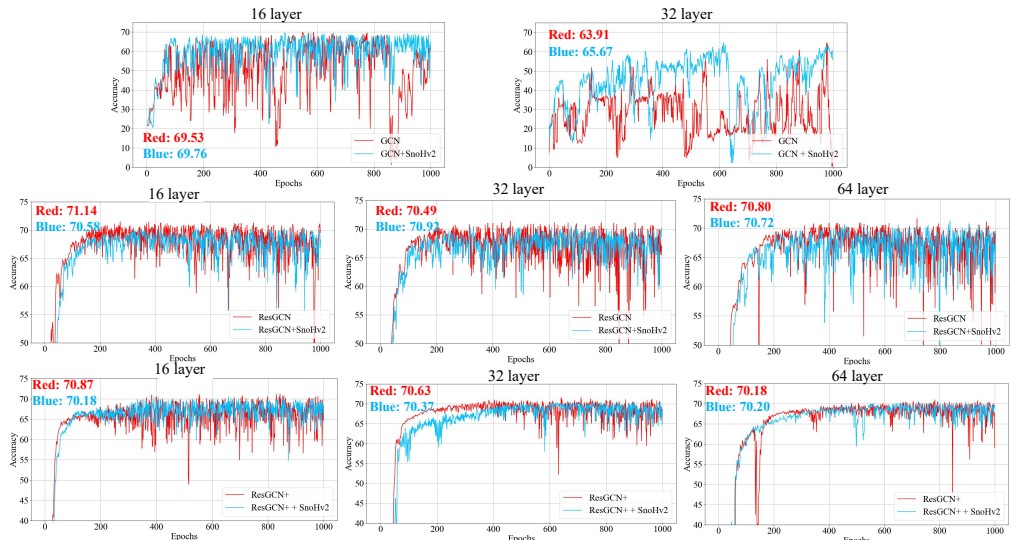

**Figure 5:** Performance comparisons on 16, 32, 64 layers using SnoHv2 across GCN, ResGCN and ResGCN+.

**Table 4:** Comparison between different backbones with SnoHv1 on Ogbn-Arxiv, where L denotes layers.

| | **GCN**, $\rho = 0.08, 0.05$ | | | **ResGCN**, $\rho = 0.1, 0.07, 0.02$ | | | **ResGCN+**, $\rho = 0.1, 0.07, 0.02$ | | |
|---|---|---|---|---|---|---|---|---|---|
| | **16-L** | **32-L** | **64-L** | **16-L** | **32-L** | **64-L** | **16-L** | **32-L** | **64-L** |
| +SnoHv1 | 69.89 | 55.88 | Collapse | 71.78 | 71.17 | 70.96 | 70.01 | 70.93 | 70.44 |
| w/o SnoHv1 | 69.47 | 53.84 | Collapse | 70.53 | 70.43 | 70.61 | 70.79 | 70.58 | 70.17 |

at which it finds the winning ticket, and compare it to the optimal sparsity of SnoHv1/v2. We control the pruning rate of each layer in SnoHv1 to be 0.3, while in SnoHv2, we set $\rho$ to 0.2. All network training for 1000 epochs with learning rate 0.001 with Adam optimizer. As shown in Fig 6, we find that our pruning rate on the ResGCN is higher than that of the graph lottery ticket, yet we can achieve relatively comparable performance. This corroborates the possibility that deep layer aggregation may indeed no longer contribute to the model, allowing for stopping at shallower layers.

## 4.2 SNOHV2 ON OGBN-PROTEIN AND OGBN-PRODUCT

To verify the scalability of our model on large datasets, we further expanded the dataset size and tested its performance on datasets with tens of millions of edges, by employing the Ogbn-Protein and Ogbn-Product datasets as benchmarks. Due to the high complexity of training on the entire graph, we adopted the common subgraph training approach (Chiang et al., 2019). As our pruning mask is static, we split the above two datasets into a fixed number of subgraphs for training (we set values as 30, 10 for two graphs in this work). Based on the aforementioned algorithms, we integrated GCN, ResGCN, and ResGCN+, referring to them as Cluster-GCN, Cluster-Res and Cluster-Res+ respectively. Due to the need to measure an excessive number of edge element gradients, the implementation efficiency of SnoHv1 on these two datasets is relatively low. Therefore, we only use SnoHv2 as our method for hypothesis verification.

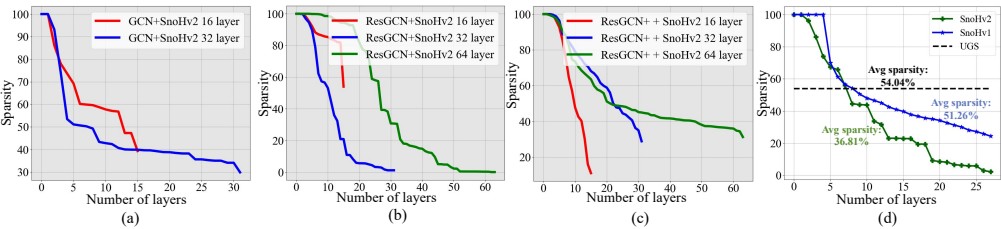

**Figure 6:** (a) (b) (c) denote the edge sparsity under different backbones. It is worth noting that sparsity is represented as the ratio of the remaining edges to the total number of edges. (d) represents the sparsity of each layer under ResGCN+Arxiv setting of SnoHv1/v2 and UGS.

**Table 5:** Comparisons with SnoHv2 on Ogbn-Proteins/Products, where L denotes layers. w/o denotes without.

|  | Cluster-GCN, $\rho = 0.15$ | | | Cluster-Res, $\rho = 0.15$ | | | Cluster-Res+, $\rho = 0.15$ | | |
| --- | --- | --- | --- | --- | --- | --- | --- | --- | --- |
|  | 16-L | 32-L | 64-L | 16-L | 32-L | 64-L | 16-L | 32-L | 64-L |
| Ogbn-Proteins | 71.88 | 71.32 | 71.08 | 79.80 | 78.87 | OOM | 80.04 | 79.32 | OOM |
| Ogbn-Proteins (w/o) | 71.32 | 68.44 | 70.55 | 78.40 | 77.71 | OOM | 79.90 | 79.05 | OOM |
| Ogbn-Product | 68.46 | 69.44 | OOM | 72.17 | 72.22 | OOM | 72.34 | 72.64 | OOM |
| Ogbn-Product (w/o) | 68.40 | 69.18 | OOM | 71.45 | 70.94 | OOM | 71.27 | 71.09 | OOM |

As depicted in Table 5, the adoption of SnoHv2 leads to a considerable improvement when compared to conventional backbones. These findings align with the observed behavior in smaller datasets. Specifically, under the configuration of 16 and 32 layers with the combination of protein and Cluster-Res, we manage to outperform the baseline by approximately 1.0%. On the GCN architecture, a notable enhancement of almost 3.0% was achieved at 32 layers. This further clarifies the validity of our hypothesis. Intriguingly, we uncover that denser graphs, such as Ogbn-Proteins, demonstrate greater resilience to sparsification. Upon contrasting the node classification outcomes on Ogbn-ArXiv (average degree$\approx$13.77) and Ogbn-Proteins (average degree$\approx$597.00), it becomes evident that Ogbn-Proteins maintains only a minimal performance discrepancy with SnoHv2, even when applied to heavily pruned graphs ($\approx$ 34.77%, sparsity of SnoHv2+Arxix $\approx$ 36.81%), this finding is also consistent with the conclusions drawn in (Chen et al., 2020c).

## 5 CASE STUDY

After investigating the overall performance of the SnoH in Sec 3 and 4, we further turn to qualitatively analyze the effect of the receptive fields on certain nodes in Citeseer graph via some case studies shown in Fig 7, where all the accuracy scores belonging to the same settings and GCN baseline. Based on the information conveyed in Fig 7, the following observations can be made: **Obs 1 (top line).** Generally, we argue that nodes with a greater number of neighbors should be assigned a relatively higher pruning rate, while those with fewer neighboring information should

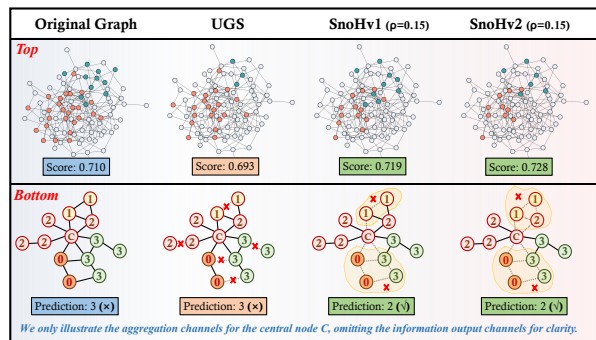

**Figure 7:** *Top.* Illustration of the two-hop receptive fields for two nodes (blue and red) along with the results showcased by different algorithms. *Bottom.* Prediction results for the central node C (The label is 2) using different algorithms.

be inclined to be retained. However, UGS does not adequately preserve the adjacency matrices of nodes with fewer neighbors within the receptive field, which can be detrimental to the prediction of certain nodes. The SnoH, in comparison to UGS, might exhibit better selectivity in this aspect. This ensures predictive capability for certain nodes and overcomes over-smoothing issues. **Obs 2 (bottom line).** When closely examining a particular node, it becomes evident that obstructing specific information transmission channels to the central node fosters a heightened focus on its pivotal neighbors. This, in turn, culminates in a more precise final prediction. The comprehensive blocking of entire aggregation channels for less critical neighbors underscores how SnoH effectively mitigates the challenges associated with overfitting and over-smoothing.

## 6 CONCLUSION & FUTURE WORK

In this paper, we have presented the Snowflake Hypothesis for the first time to discover the unique receptive field of each node, carefully suggesting the depth of early stopping for each node through the prevalent techniques of adjacency matrix pruning and cosine distance judgment. Our experiments on multiple graph datasets have demonstrated that early stopping of node aggregation at different depths can effectively enhance inference efficiency (pruning benefits), overcome the over-smoothing problem (early stopping benefits), and simultaneously offer better interpretability. Our framework is both general and succinct, compatible with many mainstream deep networks, such as ResGCN, JKNet, *etc.*, to boost performance and can also be integrated with different training strategies. Our empirical study of the existence of snowflakes invites a number of future work and research questions. We have listed the potential research points and future work in Appendix H.

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

## A    DETAILS OF SNOHV1

In this section, we start by providing a simple illustration of our second version model, SnoHv1, through examples. We then delve into the detailed description of our algorithmic workflow. As shown in the Fig 8, we execute our SnoHv1 on a three-layer GCN. The adjacency matrix undergoes an assigning process to align the pruning elements from inner layers with the outer layer's adjacency matrix, propagating their influence layer by layer. Additionally, the outer layer also adds pruning edges during each individual pruning, ensuring that the same node has different aggregation depths on different neighbors. Compared to SnoHv2, this type of pruning allows for more refined handling of the node's receptive field issue.

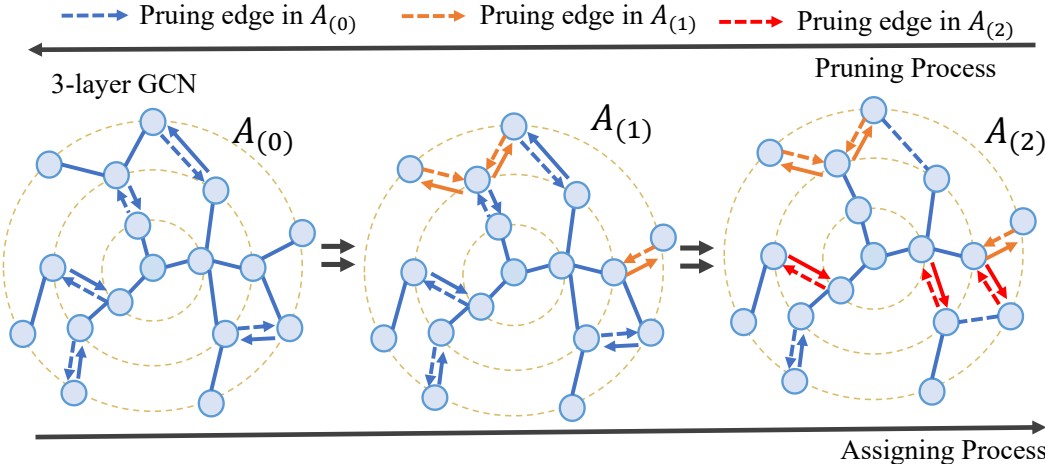

**Figure 8:** An example of our SnoHv1 in 3-layer GCN.

## B    TRAINING SCHEMES

In this part, we present the detailed processes of the three training methods in the Fig 9. For SnoHv1/v2(O), we employ the one-shot pruning training approach, where a complete pruning of the adjacency matrix is performed every $k$ iterations, removing p% of the rows. This process is repeated until all adjacency matrices are traversed. For SnoHv1/v2(IP), we refine the pruning of an adjacency matrix into an iterative pruning process, where a portion of rows (v1) or elements (v2) are pruned at each regular iterations, followed by continued training and multiple pruning iterations. It is worth noting that our approach resembles the UGS algorithm (Chen et al., 2020c), however, the key difference lies in our pruning being based on the notion of receptive fields. The inner layer's adjacency matrix influences the size of the receptive field in the outer layer, providing better interpretability and algorithmic rationality in an intuitive sense. As for SnoHv1/v2(ReI), during the training process, it is possible that while adjusting the receptive fields of the outer layer's adjacency matrix, the network parameters have already converged to a relatively good local optima. At this point, pruning the inner layer's adjacency matrix may have minimal impact on the model's performance. To address this, we employ a re-initialization (ReI) strategy. After each adjacency matrix pruning is completed, we re-initialize the entire model while keeping the pruned adjacency matrix fixed. Subsequently, we proceed to train and optimize the inner layer's adjacency matrix. Although the training processes of SnoHv1/v2(ReI) and SnoHv1/v2(O) involve $k$ iterations, they may not necessarily be the same in practice. For ease of representation, we use $k$ to denote the number of training iterations. However, in the actual implementation, we will provide specific values for the hyperparameters.

## C    DETAILS OF DATASETS AND BACKBONES

In this section, we provide detailed descriptions of the datasets used in this paper. The statistical characteristics of the datasets are presented in Table 6.

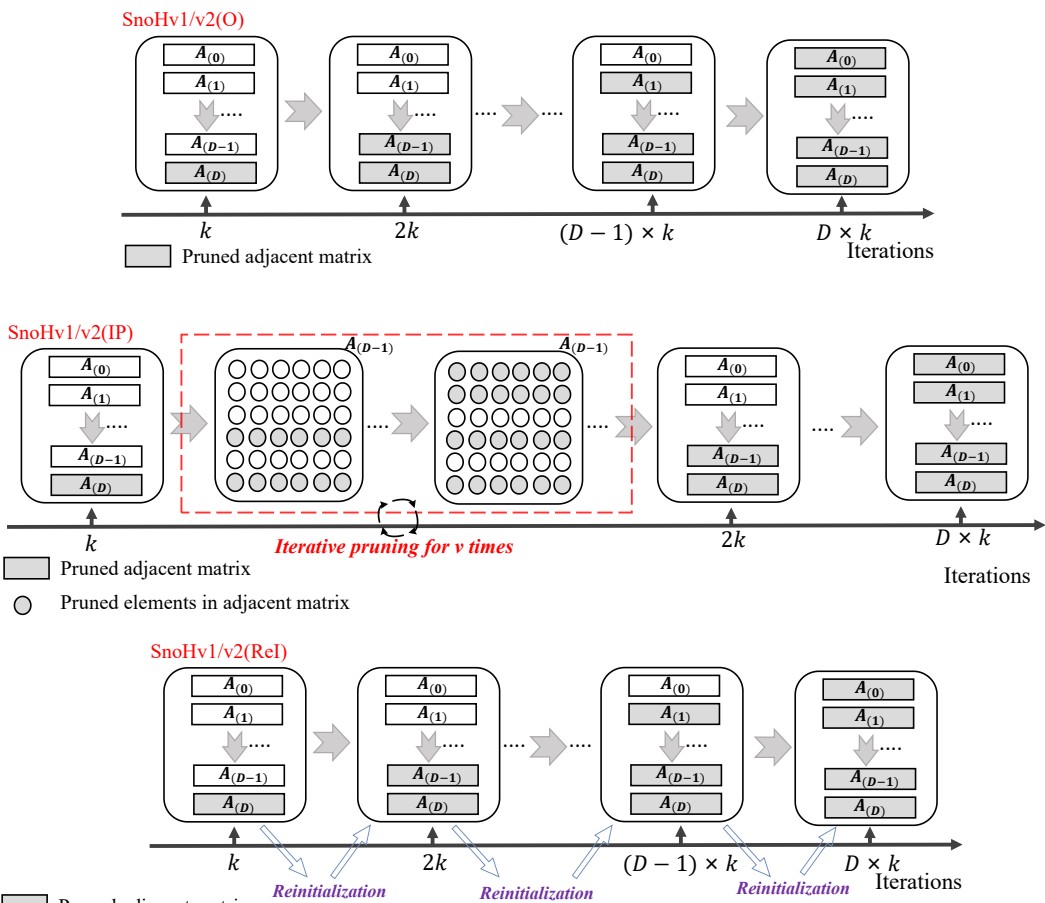

**Figure 9:** An illustration of three training schemes depicted in our paper.

**Table 6:** Statistical characteristics of the dataset used in our paper.

| Dataset | Task | #Nodes | #Edges | #Classes | Evaluation Metric |
|---------|------|--------|--------|----------|-------------------|
| Cora | Node classification | 2,708 | 5,278 | Multi-class | Accuracy |
| Citeseer | Node classification | 3,327 | 4,732 | Multi-class | Accuracy |
| PubMed | Node classification | 19,717 | 88,338 | Multi-class | Accuracy |
| Ogbn-Proteins | Node classification | 132,534 | 39,561,252 | Binary | ROC-AUC |
| Ogbn-Products | Node classification | 2,449,029 | 61,859,140 | Multi-class | Accuracy |
| Ogbn-Arxiv | Node classification | 169,343 | 1,166,243 | Multi-class | Accuracy |

# D  RELATED WORK

**Graph Neural Networks**. The Graph Neural Networks (GNNs) family encompasses a diverse array of graph message-passing architectures, each capable of integrating topological structures and node features to create more expressive representations of the entire graph. The efficacy of GNNs, as we illustrate, primarily originates from their inherent "message-passing" function, represented as $H^{(k)} = M\left(A, H^{(k-1)}; \Theta^{(k)}\right)$. Here, $H^{(k)} \in \mathbb{R}^{|\mathcal{V}| \times F}$ corresponds to the node embedding after $k$ iterations of GNN aggregation, $M$ denotes the message propagation function, and $\Theta^{(k)}$ signifies the trainable parameters at the $k$-th layer of the GNN ($H^{(0)} = X$). In light of the growing popularity of graph neural networks, a myriad of propagation functions $M$ (Gilmer et al., 2017; Hamilton et al., 2017) and GNN variants (Estrach et al., 2014; Velickovic et al., 2017; Li et al., 2020; Mavromatis & Karypis, 2020) have emerged.

**Deep Graph Neural Networks.** Despite the promising results obtained by GNNs, they encounter notorious over-smoothing and overfitting issues when scaling up to deep structure. To this end, many streams of work have been dedicated to solving these issues and help GNNs have a deep structure. A prominent approach is to inherit the depth modules of CNNs to the graph realm, such as skip and residual connections (Li et al., 2019; Sun et al., 2019; Xu et al., 2018; Li et al., 2021; Xu et al., 2018; Chen et al., 2020b; Xu et al., 2021). However, these works do not involve customized operations for the receptive field of each node and lack a specific understanding of graphs. Another representative is combine deep aggregation strategies with shallow GNNs (Wu et al., 2019; Chien et al., 2020; Liu et al., 2020; Zou et al., 2019; Rong et al., 2019; Gasteiger et al., 2019; Huang et al., 2023; Wang et al., 2022a). Similarly, these works prevent the over-smoothing issue by replacing the aggregation strategy of the entire network, lacking an understanding of node-specific differentiations. There are also some works that make efforts to theoretically propose methods for training deep GNNs (Xu et al., 2021; Min et al., 2020). However, these works are limited to specific types of GNNs, lacking generalizability and practical significance.

**Graph Pooling & Sampling.** Graph pooling and sampling devote to reducing the computational burden of GNNs by selectively sampling sub-graphs or applying pruning methods (Chen et al., 2018; Eden et al., 2018; Chen et al., 2021; Eden et al., 2018; Chen et al., 2021; Gao & Ji, 2019; Lee et al., 2019). We divide current graph pooling or sampling techniques into two categories. (1) *Sampling-based methods* aims at selecting the most expressive nodes or edges (i.e., dropping the rest) from the original graph to construct a new subgraph (Gao & Ji, 2019; Lee et al., 2019; Ranjan et al., 2020; Zhang et al., 2021). Though efficient, the dropping of nodes/edges sometimes results in severe information loss and isolated subgraphs, which may cripple the performance of GNNs (Wu et al., 2022). (2) *Clustering-based methods* learns how to cluster together nodes in the original graph, and produces a reduced graph where the clusters are set as nodes (Ying et al., 2018; Wu et al., 2022; Roy et al., 2021), which can remedy the aforementioned information loss problem.

**Lottery Ticket Hypothesis (LTH).** LTH articulates that a sparse and admirable subnetwork can be identified from a dense network by iterative pruning (Frankle & Carbin, 2018). LTH is initially observed in dense networks and is broadly found in many fields (Evci et al., 2020; Frankle et al., 2020; Malach et al., 2020; Ding et al., 2021; Chen et al., 2020c; 2021). Derivative theories (Chen et al., 2020d; You et al., 2021; Ma et al., 2021) are proposed to optimize the procedure of network sparsification and pruning. In addition to them, Dual Lottery Ticket Hypothesis (DLTH) considers a more general case to uncover the relationship between a dense network and its sparse counterparts (Bai et al., 2022; Wang et al., 2022b). Recenlty, graph lottery ticket (Chen et al., 2020c) proposes to use iterative pruning methods on adjacency matrix and weights (called UGS approach) can obtain a graph lottery ticket during the trianing phase. Meanwhile, the strong potential of the LTH has been proven to promote the development of explainable AI (XAI) (Fang et al., 2023).

## E    EXPERIMENTAL SETTINGS AND RESULTS ON SMALL GRAPHS

**Experimental settings.** As for three small-scale graphs, we adopt the supervised node classification setting. In our implementation, we choose 60%, 20%, 20% split ratio as our train-val-test splitting of datasets. During the training phase, we choose Adam as optimizer and set learning rate as 0.01, and hidden layer deimension as 64. Tab 9 illustrates the experimental details of three compact datasets: Cora, Citeseer, and PubMed. In this table, the term "SnoHv1(O) PE" signifies the pruning epoch within the one-shot pruning strategy under SnoHv1, detailing the number of epochs completed prior to pruning the adjacency matrix for each layer. "SnoHv1(ReI) PE" signifies the epoch count for reinitialization under the reinitialization scenario. The symbol $\rho$ is indicative of the depth at which the model halts aggregation under SnoHv2, which can be interpreted as an early termination when the cosine distance between the consolidated output and the initial layer output falls beneath $\rho$. As a rule of thumb, a larger $\rho$ induces earlier termination at lesser depths. In our experiment, different depths may correspond to different values of $\rho$. We will later discuss in detail how the settings of $\rho$ values affect the performance of the model.

**Table 7:** Node sparsity (NS) and edge sparsity (ES) of each layer when GCN+SnoHv2 achieves optimal performance under three small datasets. Li denotes L-th layer.

| SnoHv2 32-layer GCN $\rho$=0.001 | | | | | |
|---|---|---|---|---|---|
| GCN+Cora | GCN+Citeseer | GCN+PubMed | GCN+Cora | GCN+Citeseer | GCN+PubMed |
| L0 NS: 100.00% | L0 NS: 100.00% | L0 NS: 100.00% | L0 ES: 100.00% | L0 ES: 100.00% | L0 ES: 100.00% |
| L1 NS: 100.00% | L1 NS: 100.00% | L1 NS: 100.00% | L1 ES: 100.00% | L1 ES: 100.00% | L1 ES: 100.00% |
| L2 NS: 66.32% | L2 NS: 67.06% | L2 NS: 94.90% | L2 ES: 73.86% | L2 ES: 78.79% | L2 ES: 90.66% |
| L3 NS: 43.24% | L3 NS: 41.90% | L3 NS: 85.85% | L3 ES: 50.79% | L3 ES: 50.60% | L3 ES: 78.59% |
| L4 NS: 30.39% | L4 NS: 30.69% | L4 NS: 74.99% | L4 ES: 35.32% | L4 ES: 39.25% | L4 ES: 65.41% |
| L5 NS: 16.47% | L5 NS: 20.53% | L5 NS: 71.90% | L5 ES: 22.00% | L5 ES: 25.37% | L5 ES: 61.13% |
| L6 NS: 14.11% | L6 NS: 17.79% | L6 NS: 67.93% | L6 ES: 17.49% | L6 ES: 22.76% | L6 ES: 55.95% |
| L7 NS: 12.63% | L7 NS: 17.13% | L7 NS: 61.69% | L7 ES: 15.46% | L7 ES: 22.29% | L7 ES: 48.10% |
| L8 NS: 9.12% | L8 NS: 14.70% | L8 NS: 59.53% | L8 ES: 12.25% | L8 ES: 18.38% | L8 ES: 46.21% |
| L9 NS: 8.42% | L9 NS: 14.25% | L9 NS: 59.23% | L9 ES: 11.72% | L9 ES: 17.99% | L9 ES: 45.63% |
| L10 NS: 8.42% | L10 NS: 13.98% | L10 NS: 54.65% | L10 ES: 11.72% | L10 ES: 17.42% | L10 ES: 41.52% |
| L11 NS: 3.21% | L11 NS: 10.43% | L11 NS: 54.31% | L11 ES: 6.24% | L11 ES: 13.64% | L11 ES: 40.78% |
| L12 NS: 3.06% | L12 NS: 10.10% | L12 NS: 54.13% | L12 ES: 6.19% | L12 ES: 13.15% | L12 ES: 40.65% |
| L13 NS: 2.95% | L13 NS: 9.65% | L13 NS: 53.77% | L13 ES: 6.12% | L13 ES: 12.45% | L13 ES: 40.30% |
| L14 NS: 2.66% | L14 NS: 9.56% | L14 NS: 52.41% | L14 ES: 6.00% | L14 ES: 12.41% | L14 ES: 37.90% |
| L15 NS: 2.66% | L15 NS: 7.15% | L15 NS: 51.70% | L15 ES: 6.00% | L15 ES: 9.87% | L15 ES: 37.45% |
| L16 NS: 2.55% | L16 NS: 6.88% | L16 NS: 49.26% | L16 ES: 5.90% | L16 ES: 9.70% | L16 ES: 35.97% |
| L17 NS: 2.29% | L17 NS: 6.88% | L17 NS: 49.09% | L17 ES: 5.63% | L17 ES: 9.70% | L17 ES: 35.59% |
| L18 NS: 2.03% | L18 NS: 5.77% | L18 NS: 48.92% | L18 ES: 5.49% | L18 ES: 8.07% | L18 ES: 35.36% |
| L19 NS: 1.88% | L19 NS: 5.77% | L19 NS: 47.60% | L19 ES: 5.07% | L19 ES: 8.07% | L19 ES: 34.68% |
| L20 NS: 1.14% | L20 NS: 5.53% | L20 NS: 47.49% | L20 ES: 4.23% | L20 ES: 7.88% | L20 ES: 34.58% |
| L21 NS: 1.00% | L21 NS: 5.50% | L21 NS: 45.56% | L21 ES: 3.95% | L21 ES: 7.83% | L21 ES: 32.89% |
| L22 NS: 0.89% | L22 NS: 4.84% | L22 NS: 44.47% | L22 ES: 3.89% | L22 ES: 6.05% | L22 ES: 31.81% |
| L23 NS: 0.89% | L23 NS: 4.57% | L23 NS: 44.02% | L23 ES: 3.89% | L23 ES: 5.81% | L23 ES: 31.40% |
| L24 NS: 0.89% | L24 NS: 3.91% | L24 NS: 43.86% | L24 ES: 3.89% | L24 ES: 5.05% | L24 ES: 31.28% |
| L25 NS: 0.89% | L25 NS: 3.91% | L25 NS: 43.53% | L25 ES: 3.89% | L25 ES: 5.05% | L25 ES: 30.92% |
| L26 NS: 0.81% | L26 NS: 3.85% | L26 NS: 43.33% | L26 ES: 3.46% | L26 ES: 5.00% | L26 ES: 30.82% |
| L27 NS: 0.55% | L27 NS: 3.22% | L27 NS: 43.28% | L27 ES: 2.05% | L27 ES: 3.23% | L27 ES: 30.81% |
| L28 NS: 0.52% | L28 NS: 3.01% | L28 NS: 41.57% | L28 ES: 2.04% | L28 ES: 2.93% | L28 ES: 30.25% |
| L29 NS: 0.52% | L29 NS: 2.98% | L29 NS: 41.40% | L29 ES: 2.04% | L29 ES: 2.91% | L29 ES: 30.16% |
| L30 NS: 0.48% | L30 NS: 2.98% | L30 NS: 41.37% | L30 ES: 1.99% | L30 ES: 2.91% | L30 ES: 30.15% |
| L31 NS: 0.48% | L31 NS: 2.89% | L31 NS: 27.80% | L31 ES: 1.99% | L31 ES: 2.76% | L31 ES: 17.68% |

**Table 8:** Node sparsity (NS) and edge sparsity (ES) of each layer when GCN+SnoHv2 achieves optimal performance under Cora datasets. Li denotes L-th layer.

| SnoHv2 64-layer GCN+Cora $\rho$=0.001 | |
| --- | --- |
| L17 NS: 36.93% | L17 ES: 47.08% |
| L18 NS: 36.89% | L18 ES: 47.06% |
| L19 NS: 36.89% | L19 ES: 47.06% |
| L20 NS: 36.89% | L20 ES: 47.06% |
| L21 NS: 36.41% | L21 ES: 46.82% |
| L22 NS: 36.41% | L22 ES: 46.82% |
| L23 NS: 36.41% | L23 ES: 46.82% |
| L24 NS: 36.41% | L24 ES: 46.82% |
| L25 NS: 36.37% | L25 ES: 46.80% |
| L26 NS: 36.37% | L26 ES: 46.80% |
| L27 NS: 36.37% | L27 ES: 46.80% |
| L28 NS: 36.37% | L28 ES: 46.80% |
| L29 NS: 29.21% | L29 ES: 41.43% |
| L30 NS: 25.26% | L30 ES: 37.16% |
| L31 NS: 25.26% | L31 ES: 37.16% |
| L32 NS: 25.26% | L32 ES: 37.16% |
| L33 NS: 25.26% | L33 ES: 37.16% |
| L34 NS: 25.26% | L34 ES: 37.16% |
| L35 NS: 25.26% | L35 ES: 37.16% |
| L36 NS: 25.26% | L36 ES: 37.16% |
| L37 NS: 11.82% | L37 ES: 21.92% |
| L38 NS: 11.82% | L38 ES: 21.92% |
| L39 NS: 7.87% | L39 ES: 16.89% |
| L40 NS: 7.87% | L40 ES: 16.89% |
| L41 NS: 7.87% | L41 ES: 16.89% |
| L42 NS: 7.87% | L42 ES: 16.89% |
| L43 NS: 7.72% | L43 ES: 16.70% |
| L44 NS: 7.72% | L44 ES: 16.70% |
| L45 NS: 7.53% | L45 ES: 16.36% |
| L46 NS: 7.53% | L46 ES: 16.36% |
| L47 NS: 7.53% | L47 ES: 16.36% |
| L48 NS: 7.53% | L48 ES: 16.36% |
| L49 NS: 7.53% | L49 ES: 16.36% |
| L50 NS: 7.53% | L50 ES: 16.36% |
| L51 NS: 7.53% | L51 ES: 16.36% |
| L52 NS: 7.53% | L52 ES: 16.36% |
| L53 NS: 7.53% | L53 ES: 16.36% |
| L54 NS: 7.53% | L54 ES: 16.36% |
| L55 NS: 7.53% | L55 ES: 16.36% |
| L56 NS: 7.53% | L56 ES: 16.36% |
| L57 NS: 7.53% | L57 ES: 16.36% |
| L58 NS: 7.50% | L58 ES: 16.32% |
| L59 NS: 6.79% | L59 ES: 15.54% |
| L60 NS: 6.79% | L60 ES: 15.54% |
| L61 NS: 6.79% | L61 ES: 15.54% |
| L62 NS: 6.79% | L62 ES: 15.54% |
| L63 NS: 6.57% | L63 ES: 15.26% |

**Table 9:** Implementation details of SnoH on node classification on Cora, Citeseer and PubMed datasets.

| Task | Node classification | | | | | |
| --- | --- | --- | --- | --- | --- | --- |
| Dataset | Learning rate | Optimizer | SnoHv1(O) PE | SnoHv1(ReI) PE | $\rho$ | Total training epoch |
| Cora | 0.01 | Adam | 30 | 300 | - | 1000 |
| Citeseer | 0.01 | Adam | 30 | 300 | - | 1000 |
| PubMed | 0.01 | Adam | 30 | 300 | - | 1000 |
| Ogbn-Arxiv | 0.001 | Adam | 30 | - | - | 500 |
| Ogbn-Protein | 0.01 | Adam | - | - | - | 75 |
| Ogbn-Product | 0.001 | Adam | - | - | - | 500 |

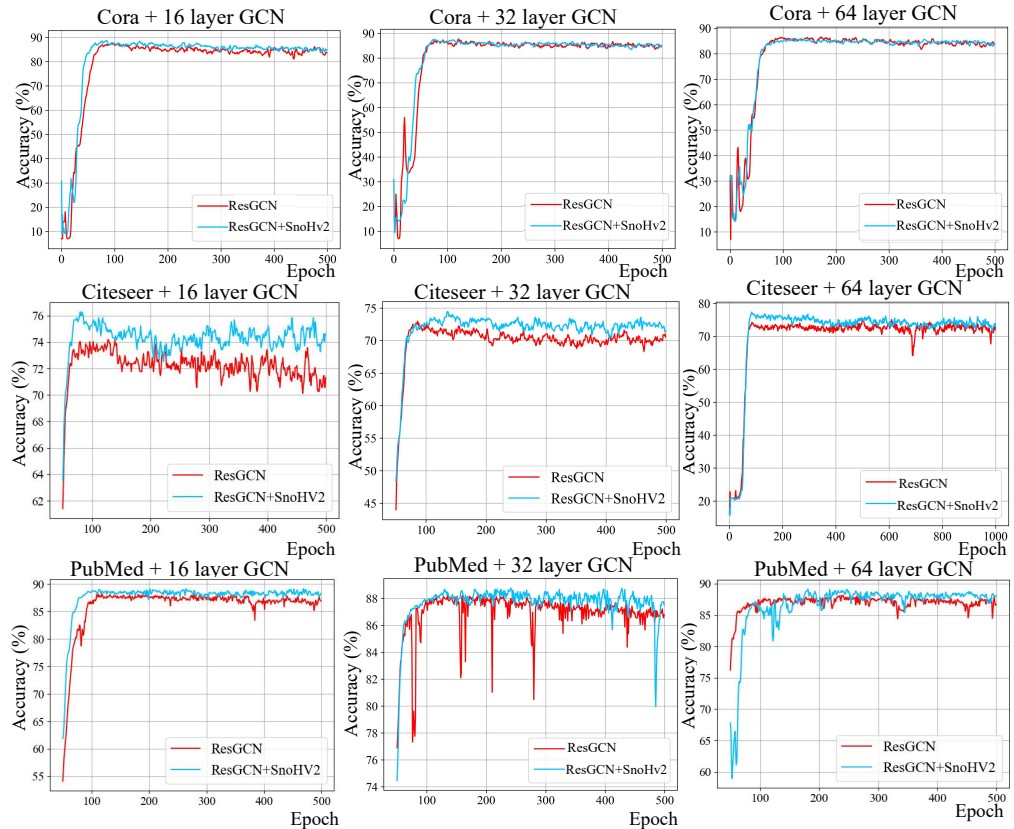

**Figure 10:** Test curves during training on 16, 32, 64 layer settings using SnoHv2 across three small graphs with ResGCN backbone. We recorded the ResGCN original performance as 85.21%, 71.65%, 87.01% on Cora, Citeseer and PubMed at 64 layers. When combined with SnoHv2, the performance is 85.90%, 72.94%, 88.11%.

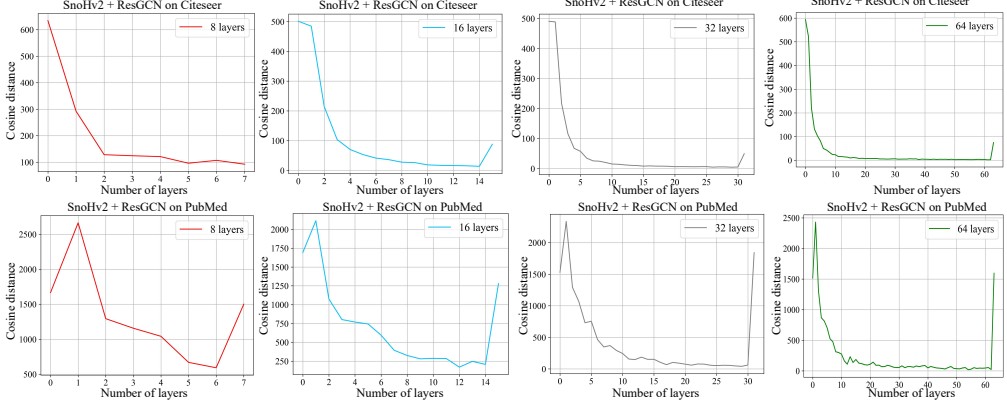

**Figure 11:** The experimental settings of ResGCN+SnoHv2 on the Citeseer and PubMed datasets are demonstrated using cosine distance. It can be clearly observed that a gradual decrease in cosine distance occurs across results obtained with 8 to 64 layers, indicating that as the depth of the GNN increases, the model exhibits oversmoothing phenomenon. Our approach effectively demonstrates this process.

**Table 10:** Edge sparsity (ES) of each layer when ResGCN+SnoHv2 or JKNet+SnoHv2 achieves optimal performance under Cora, Citeseer and PubMed datasets on 32 layer settings. Li denotes L-th layer.

| SnoHv2 32-L ResGCN/JKNet | | | | | |
|---|---|---|---|---|---|
| ResGCN+Cora | ResGCN+Citeseer | ResGCN+PubMed | JKNet+Cora | JKNet+Citeseer | JKNet+PubMed |
| L0 ES: 100.00% | L0 ES: 100.00% | L0 ES: 100.00% | L0 ES: 100.00% | L0 ES: 100.00% | L0 ES: 100.00% |
| L1 ES: 92.75% | L1 ES: 100.00% | L1 ES: 100.00% | L1 ES: 94.48% | L1 ES: 89.07% | L1 ES: 97.15% |
| L2 ES: 54.58% | L2 ES: 95.17% | L2 ES: 79.52% | L2 ES: 80.03% | L2 ES: 84.46% | L2 ES: 90.99% |
| L3 ES: 26.89% | L3 ES: 59.25% | L3 ES: 49.29% | L3 ES: 65.13% | L3 ES: 73.92% | L3 ES: 85.81% |
| L4 ES: 10.33% | L4 ES: 10.87% | L4 ES: 30.32% | L4 ES: 48.10% | L4 ES: 55.69% | L4 ES: 74.01% |
| L5 ES: 5.14% | L5 ES: 5.67% | L5 ES: 16.31% | L5 ES: 43.79% | L5 ES: 37.75% | L5 ES: 52.65% |
| L6 ES: 2.91% | L6 ES: 4.57% | L6 ES: 15.19% | L6 ES: 24.47% | L6 ES: 30.84% | L6 ES: 46.98% |
| L7 ES: 1.95% | L7 ES: 1.26% | L7 ES: 10.53% | L7 ES: 19.51% | L7 ES: 20.73% | L7 ES: 13.83% |
| L8 ES: 1.42% | L8 ES: 0.36% | L8 ES: 9.00% | L8 ES: 16.45% | L8 ES: 12.01% | L8 ES: 13.74% |
| L9 ES: 0.96% | L9 ES: 0.31% | L9 ES: 8.92% | L9 ES: 8.76% | L9 ES: 5.01% | L9 ES: 13.74% |
| L10 ES: 0.86% | L10 ES: 0.31% | L10 ES: 8.57% | L10 ES: 8.72% | L10 ES: 0.00% | L10 ES: 13.74% |
| L11 ES: 0.62% | L11 ES: 0.30% | L11 ES: 8.14% | L11 ES: 8.28% | L11 ES: 0.00% | L11 ES: 13.74% |
| L12 ES: 0.57% | L12 ES: 0.27% | L12 ES: 6.27% | L12 ES: 7.78% | L12 ES: 0.00% | L12 ES: 13.74% |
| L13 ES: 0.50% | L13 ES: 0.26% | L13 ES: 5.56% | L13 ES: 0.00% | L13 ES: 0.00% | L13 ES: 0.00% |
| L14 ES: 0.46% | L14 ES: 0.18% | L14 ES: 4.14% | L14 ES: 0.00% | L14 ES: 0.00% | L14 ES: 0.00% |
| L15 ES: 0.36% | L15 ES: 0.15% | L15 ES: 2.67% | L15 ES: 0.00% | L15 ES: 0.00% | L15 ES: 0.00% |
| L16 ES: 0.30% | L16 ES: 0.14% | L16 ES: 2.38% | L16 ES: 0.00% | L16 ES: 0.00% | L16 ES: 0.00% |
| L17 ES: 0.28% | L17 ES: 0.04% | L17 ES: 2.36% | L17 ES: 0.00% | L17 ES: 0.00% | L17 ES: 0.00% |
| L18 ES: 0.27% | L18 ES: 0.04% | L18 ES: 2.22% | L18 ES: 0.00% | L18 ES: 0.00% | L18 ES: 0.00% |
| L19 ES: 0.09% | L19 ES: 0.03% | L19 ES: 1.65% | L19 ES: 0.00% | L19 ES: 0.00% | L19 ES: 0.00% |
| L20 ES: 0.09% | L20 ES: 0.03% | L20 ES: 1.65% | L20 ES: 0.00% | L20 ES: 0.00% | L20 ES: 0.00% |
| L21 ES: 0.09% | L21 ES: 0.03% | L21 ES: 1.58% | L21 ES: 0.00% | L21 ES: 0.00% | L21 ES: 0.00% |
| L22 ES: 0.09% | L22 ES: 0.01% | L22 ES: 1.55% | L22 ES: 0.00% | L22 ES: 0.00% | L22 ES: 0.00% |
| L23 ES: 0.07% | L23 ES: 0.01% | L23 ES: 1.34% | L23 ES: 0.00% | L23 ES: 0.00% | L23 ES: 0.00% |
| L24 ES: 0.07% | L24 ES: 0.01% | L24 ES: 1.22% | L24 ES: 0.00% | L24 ES: 0.00% | L24 ES: 0.00% |
| L25 ES: 0.07% | L25 ES: 0.01% | L25 ES: 1.08% | L25 ES: 0.00% | L25 ES: 0.00% | L25 ES: 0.00% |
| L26 ES: 0.07% | L26 ES: 0.00% | L26 ES: 0.90% | L26 ES: 0.00% | L26 ES: 0.00% | L26 ES: 0.00% |
| L27 ES: 0.07% | L27 ES: 0.00% | L27 ES: 0.90% | L27 ES: 0.00% | L27 ES: 0.00% | L27 ES: 0.00% |
| L28 ES: 0.05% | L28 ES: 0.00% | L28 ES: 0.80% | L28 ES: 0.00% | L28 ES: 0.00% | L28 ES: 0.00% |
| L29 ES: 0.04% | L29 ES: 0.00% | L29 ES: 0.76% | L29 ES: 0.00% | L29 ES: 0.00% | L29 ES: 0.00% |
| L30 ES: 0.04% | L30 ES: 0.00% | L30 ES: 0.61% | L30 ES: 0.00% | L30 ES: 0.00% | L30 ES: 0.00% |
| L31 ES: 0.04% | L31 ES: 0.00% | L31 ES: 0.56% | L31 ES: 0.00% | L31 ES: 0.00% | L31 ES: 0.00% |

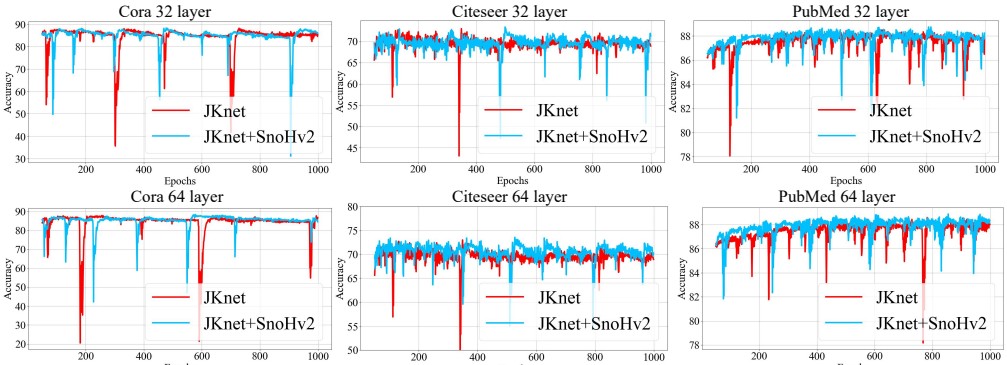

**Figure 12:** The experimental settings of JKNet+SnoHv2 on the Cora, Citeseer and PubMed datasets with 32 and 64 layers.

**Table 11:** Performance comparisons on 8, 16, 32 layer settings using SnoHv1(O), SnoHv1(IP), and SnoHv1(ReI) across three small graphs, all experimental results are the average of three runs.

| Backbone | 8 layers | | | 16 layers | | | 32 layers | | |
|---|---|---|---|---|---|---|---|---|---|
| | SnoHv1(O) | SnoHv1(IP) | SnoHv1(ReI) | SnoHv1(O) | SnoHv1(IP) | SnoHv1(ReI) | SnoHv1(O) | SnoHv1(IP) | SnoHv1(ReI) |
| *Train scheme: SnoHv1(O), Dataset: Cora, 2-layer performance: GCN without BN = 85.37* | | | | | | | | | |
| GCN | 84.37 | 83.77 | 84.30 | 83.19 | 82.21 | 82.77 | 82.09 | 82.76 | 82.45 |
| ResGCN | 85.12 | 85.47 | 84.99 | 84.35 | 85.17 | 84.72 | 85.37 | 85.21 | 85.87 |
| JKNet | 85.43 | 85.35 | 84.44 | 86.11 | 85.87 | 86.01 | 86.47 | 87.39 | 85.21 |
| *Train scheme: SnoHv1/v2(O), Dataset: Citeseer, 2-layer performance: GCN without BN = 72.44* | | | | | | | | | |
| GCN | 73.39 | 73.45 | 72.11 | 72.29 | 72.45 | 72.10 | 68.71 | 67.59 | 68.98 |
| ResGCN | 71.71 | 71.54 | 72.01 | 71.98 | 70.14 | 70.43 | 72.01 | 71.45 | 70.49 |
| JKNet | 71.34 | 70.57 | 71.38 | 70.47 | 70.98 | 71.03 | 69.89 | 68.47 | 68.47 |
| *Train scheme: SnoHv1/v2(O), Dataset: PubMed, 2-layer performance: GCN without BN = 86.50* | | | | | | | | | |
| GCN | 86.15 | 85.78 | 86.21 | 84.37 | 84.23 | 84.65 | 83.54 | 83.66 | 83.06 |
| ResGCN | 87.41 | 86.24 | 85.30 | 87.40 | 86.54 | 86.01 | 87.26 | 87.01 | 86.45 |
| JKNet | 88.29 | 87.68 | 88.11 | 87.27 | 86.57 | 85.92 | 88.65 | 87.81 | 86.53 |

**Training scheme on SnoHv1.** In this section, we further test different training strategies for SnoHv1, including one-shot pruning, iterative pruning, and re-initialization pruning strategies. (1) The commonly used strategy is one-shot pruning, where we prune the adjacency matrix of each layer during each training process. (2) The iterative pruning method involves splitting each training process and pruning some elements of the adjacency matrix in each layer during each epoch. The training continues iteratively by removing elements from the adjacency matrix. (3) The re-initialization strategy prunes one layer of the adjacency matrix at a time. We set the training epoch to 200, meaning that every 200 epochs, we determine which elements in each layer's adjacency matrix should be pruned.

We found that under different training strategies, there was no significant difference in the model's performance. All three training strategies achieved similar performance levels. However, the iterative pruning process, which involves repeatedly determining important parameters, was executed on the CPU and proved challenging to accelerate using a GPU. Additionally, when applied to large graphs, this iterative parameter evaluation process consumed a substantial amount of time. Similarly, the re-initialization method, with its repeated training and reinitialization to assess important parameters in each layer, resulted in significant time wastage. In some cases, it even took more than $D$ times the original training time ($D$ is the network depth for GNNs), which hampers its scalability to large graphs. **Based on the above observations, we conducted tests on our framework using large graphs, specifically employing the SnoHv2 version. We believe that our findings can provide valuable insights for future research in evaluating and testing various new designs.**

**The effect of $\rho$ on SnoHv2.** Interestingly, we observed varying sensitivities of the parameter $\rho$ across different datasets. The extreme sparsity of the deep adjacency matrix depends on the properties of graphs and backbones. Specifically, on sparse graphs like Cora and Citeseer, as the depth of the GCN increases, the stop rate $\rho$ gradually decreases. For example, with a 16-layer Cora+SnoHv2 configuration, the optimal value for $\rho$ is 0.4, while for 32 and 64 layers, the optimal values are 0.2 and 0.05, respectively. However, on moderately large datasets such as PubMed, sometimes a larger value of $\rho$ can lead to performance improvement.

This phenomenon also shows slight variations with different backbone architectures. When introducing residual structures, the sparsity of the deep adjacency matrix becomes even higher. This might be because residual structures preserve shallow layer information, reducing the need for deep layer information to assist in predictions.

**Compare with graph lottery tickets (UGS algorithm).** In our experiment, we compared the model performance of GCN+SnoHv2 and UGS, as well as random pruning under configurations of 8, 16, and 32 layers. We were pleasantly surprised to find that our results considerably outperformed those of random pruning and UGS, particularly under these deep-layer conditions. Our model demonstrated excellent performance. For instance, under a 16-layer setup for the Citeseer dataset, our results were 6.57% better than the best UGS configuration (five iterations of pruning, with each iteration pruning 5%). This trend was consistent across all datasets and under various depth settings, further corroborating the superior capabilities of our algorithm in deep scenarios.

Upon further analysis, we believe that our enhanced performance stems from the early stopping of the receptive field for some nodes in graphs. In fact, our network can be understood as a GCN in the shallow layers and approximates an MLP in the deeper layers. While it ceases to aggregate information for nodes in the deeper layers, it successfully circumvents the issue of gradient vanishing that often plagues deep MLPs.

**Table 12:** Comparison performances of SnoHv2 with UGS and random pruning (RP). Here IPR denotes iterative pruning rate and we set number of layers as 8. We use GCN backbone and set early stopping threshold of cosine distance as $\rho$ (Detailed descriptions in Appendix E).

| Dataset | RP | UGS(IPR=5%) | UGS(IPR=10%) | UGS(IPR=20%) | SnoHv2 | GCN |
|---|---|---|---|---|---|---|
| Cora (L=8) | 69.60 | 73.64 | 66.01 | 53.29 | 85.68 | 85.11 |
| Citeseer (L=8) | 45.50 | 65.80 | 51.50 | 43.10 | 73.24 | 72.39 |
| PubMed (L=8) | 77.82 | 84.33 | 80.91 | 71.05 | 86.56 | 86.41 |
| Cora (L=16) | 51.98 | 60.32 | 55.53 | 47.24 | 84.19 | 83.75 |
| Citeseer (L=16) | 60.36 | 66.31 | 58.12 | 30.13 | 72.33 | 71.28 |
| PubMed (L=16) | 53.22 | 79.22 | 72.52 | 58.39 | 85.79 | 84.77 |
| Cora (L=32) | 58.25 | 69.25 | 53.64 | 39.20 | 83.09 | 80.33 |
| Citeseer (L=32) | 51.95 | 57.37 | 50.31 | 51.24 | 69.89 | 68.99 |
| PubMed (L=32) | 58.32 | 77.42 | 64.26 | 60.77 | 84.06 | 83.76 |

We also evaluate the joint pruning algorithm of UGS to test our fusion capability with weight pruning on smaller graphs. We employ GCN and GAT as backbones, and conduct tests on Cora, Citeseer, and PubMed to examine the results under various depths ($4 \rightarrow 16$ layers). We control the weight sparsity from 0 to 90%, testing SnoHv1 and UGS by iteratively pruning the graph 10 times at a pruning rate of 5%, and finally showcasing the optimal results when finding GLTs at graph sparsity.

As shown in Fig 13 and 14, we find that UGS struggles to locate GLTs at different sparsity levels, in contrast, SnoHv1 manages to find a relatively favorable lottery ticket. Interestingly, we discovered that the benefits yielded from joint pruning for the SnoH algorithm even surpass those from UGS. We speculate that this might be due to our algorithm having a higher pruning rate for graphs, which further substantiates the generalization capability of our algorithm in the pruning domain. This also significantly benefits the graph lottery ticket research line. We argue that our algorithm is extremely versatile and can exhibit substantial benefits across varying network depths, backbone configurations, and pruning scenarios.

**Table 13:** The performance comparison between UGS and SnoHv1 in discovering graph lottery tickets (GLTs) on GCN backbone across various weight sparsity settings (10% → 90%) and GNN layer configurations (4 → 16 layers).

| Weight Sparsity | Method | Cora | | | | Citeseer | | | | PubMed | | | |
|---|---|---|---|---|---|---|---|---|---|---|---|---|---|
| | | 4 | 8 | 12 | 16 | 4 | 8 | 12 | 16 | 4 | 8 | 12 | 16 |
| 0% | Baseline | 83.95 | 83.65 | 84.80 | 83.60 | 75.10 | 74.20 | 73.80 | 74.00 | 88.10 | 84.40 | 85.60 | 83.10 |
| 10% | UGS | 84.05 | 83.72 | 82.43 | 81.85 | 71.45 | 71.59 | 72.80 | 75.00 | 88.73 | 86.42 | 83.77 | 80.32 |
| | SnoHv1 | 84.17 | 85.27 | 84.12 | 82.75 | 75.35 | 73.19 | 75.21 | 74.59 | 88.11 | 8.27 | 85.42 | 83.79 |
| 30% | UGS | 81.35 | 81.67 | 83.69 | 83.68 | 70.46 | 72.00 | 72.11 | 74.23 | 88.29 | 82.09 | 85.78 | 82.94 |
| | SnoHv1 | 85.64 | 85.95 | 83.78 | 81.05 | 76.35 | 74.66 | 74.89 | 72.68 | 88.71 | 86.42 | 86.18 | 83.01 |
| 50% | UGS | 78.94 | 78.97 | 80.03 | 81.65 | 68.95 | 72.20 | 71.22 | 73.39 | 86.07 | 84.57 | 82.22 | 79.50 |
| | SnoHv1 | 85.83 | 84.53 | 84.92 | 81.15 | 76.65 | 74.29 | 76.32 | 73.45 | 88.69 | 85.66 | 83.44 | 82.65 |
| 70% | UGS | 77.45 | 75.85 | 79.08 | 76.84 | 65.72 | 68.17 | 67.80 | 66.91 | 78.21 | 78.92 | 76.40 | 76.23 |
| | SnoHv1 | 84.93 | 84.00 | 80.70 | 77.15 | 75.63 | 74.29 | 73.68 | 73.10 | 88.96 | 86.74 | 86.40 | 82.75 |
| 90% | UGS | 75.35 | 70.95 | 70.51 | 67.26 | 64.55 | 62.08 | 53.20 | 58.61 | 76.40 | 76.11 | 72.27 | 70.09 |
| | SnoHv1 | 85.43 | 83.35 | 81.21 | 78.25 | 76.09 | 74.88 | 70.56 | 64.52 | 89.08 | 84.27 | 85.38 | 80.16 |

**Table 14:** The performance comparison between UGS and SnoHv1 in discovering GLTs on GAT backbone across various weight sparsity settings (10% → 90%) and GNN layer configurations (4 → 16 layers).

| Weight Sparsity | Method | Cora | | | | Citeseer | | | | PubMed | | | |
|---|---|---|---|---|---|---|---|---|---|---|---|---|---|
| | | 4 | 8 | 12 | 16 | 4 | 8 | 12 | 16 | 4 | 8 | 12 | 16 |
| 0% | Baseline | 78.20 | 78.08 | 76.12 | 75.53 | 69.82 | 67.50 | 67.40 | 67.56 | 78.10 | 76.82 | 76.30 | 76.81 |
| 10% | TGLT | 78.79 | 79.39 | 73.52 | 71.49 | 70.11 | 67.79 | 68.10 | 67.08 | 78.64 | 77.91 | 77.09 | 76.92 |
| | SnoHv1 | 79.22 | 78.28 | 76.70 | 75.31 | 69.90 | 67.69 | 68.41 | 67.82 | 78.44 | 78.22 | 77.98 | 77.10 |
| 30% | TGLT | 78.25 | 78.80 | 73.28 | 70.66 | 69.94 | 66.34 | 63.52 | 63.90 | 78.35 | 76.93 | 76.49 | 73.66 |
| | SnoHv1 | 78.71 | 78.67 | 77.10 | 76.97 | 69.83 | 67.77 | 67.54 | 68.20 | 78.64 | 78.49 | 77.04 | 76.91 |
| 50% | TGLT | 78.21 | 76.42 | 70.19 | 72.00 | 69.86 | 67.73 | 64.49 | 60.70 | 78.13 | 74.36 | 72.97 | 70.28 |
| | SnoHv1 | 78.64 | 78.35 | 76.57 | 74.83 | 69.83 | 67.46 | 68.45 | 67.07 | 78.34 | 77.12 | 76.28 | 75.33 |
| 70% | TGLT | 73.74 | 73.65 | 70.56 | 70.88 | 67.26 | 63.40 | 60.77 | 62.89 | 74.48 | 69.08 | 68.71 | 65.58 |
| | SnoHv1 | 78.65 | 77.92 | 72.65 | 70.85 | 69.90 | 67.02 | 66.38 | 66.22 | 79.12 | 77.42 | 73.08 | 72.15 |
| 90% | TGLT | 70.10 | 67.71 | 62.28 | 63.56 | 64.22 | 63.19 | 55.46 | 54.76 | 66.80 | 67.72 | 59.71 | 60.02 |
| | SnoHv1 | 74.63 | 73.28 | 68.12 | 65.78 | 63.28 | 64.30 | 61.38 | 60.23 | 76.01 | 76.88 | 69.50 | 67.42 |

# F    GENERALIZATION VALIDATION EXPERIMENTS ON GIN AND GAT

n this subsection, we evaluated the generalization ability of SnoHv2. We adopted the experimental setups of GIN+SnoHv2 and GAT+SnoHv2, and conducted comprehensive experiments on the Citeseer and Pubmed datasets. We found that, under the aforementioned experimental setups, the model almost consistently demonstrated performance improvements. Particularly, with GIN+SnoHv2 under PubMed, the model exhibited the most pronounced performance enhancement, achieving a performance improvement range of 0.22% to 3.1%. These results further clarify the excellent generalization and extensibility of the snowflake hypothesis.

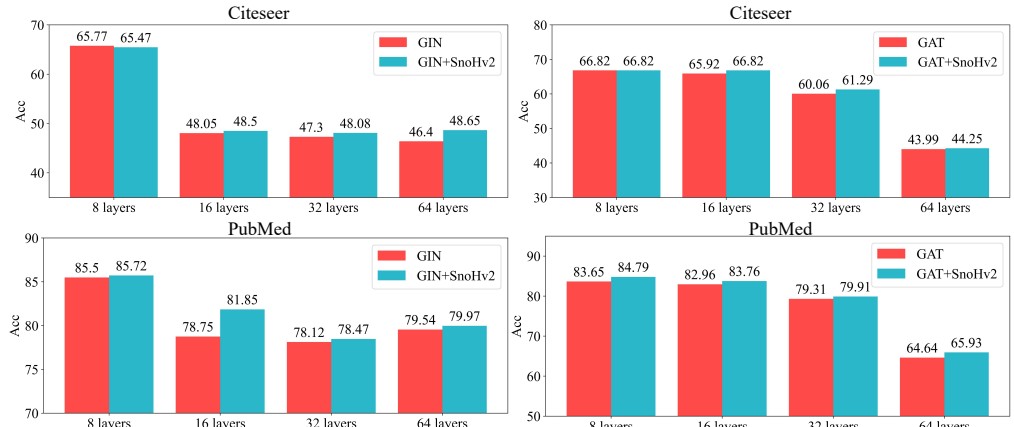

**Figure 13:** The experimental settings of GIN+SnoHv2 and GAT+SnoHv2 on the Citeseer and PubMed datasets are demonstrated using cosine distance. It can be readily observed that our algorithm significantly improves the performance of various GNN backbones.

# G    HOMOPHILY RATIO

**Definition.** Homophily indicates that adjacent nodes in the graph are likely to have similar attributes or labels. In a social network, for example, people with similar interests or beliefs tend to connect with each other. This pattern holds true in various kinds of networks, and its presence can significantly affect the way GNNs process and learn from the graph.

**Impact on GNN Learning.** In GNNs, information is often propagated between neighboring nodes, and node embeddings are updated based on the features of adjacent nodes. If the graph exhibits homophily, this propagation of information is likely to reinforce consistent features among neighboring nodes, which can make learning tasks like node classification more tractable.

**Challenges.** Conversely, if a graph does not exhibit homophily (i.e., similar nodes are not more likely to be connected), this can present challenges for learning. GNN models might have difficulty making accurate predictions or inferences in such cases, as neighboring nodes may provide conflicting or less relevant information.

**Measuring Homophily.** In some scenarios, quantifying the level of homophily can be beneficial for understanding the graph's structure and for selecting or designing appropriate models or algorithms. Various metrics and analyses might be used to gauge the extent of homophily within a given graph.

**Heterophily.** The opposite of homophily is heterophily, where neighboring nodes are more likely to be dissimilar. Recognizing whether a graph is more homophilous or heterophilous can be essential in choosing the correct approach and model for graph-based learning tasks.

In summary, homophily within GNNs signifies the inclination of connected nodes to exhibit similar attributes. This phenomenon is fundamental to the way GNNs interpret and learn from graphs, guiding not only the design but also the interpretation of various graph learning tasks. Its understanding leads to more effective model development and nuanced analysis.

$$\frac{1}{|\mathcal{V}|} \sum_{v \in \mathcal{V}} \frac{|\{(w,v) : w \in \mathcal{N}(v) \wedge y_v = y_w\}|}{|\mathcal{N}(v)|} \quad (1)$$

Through heterogeneity analysis (Pei et al., 2020), We use Eq 1 to calculate the degree of isomorphism in Arxiv, and we find that the homophily degree in Arxiv is relatively low (0.635). This might cause our SnoHv2 to be deeper in the early stopping networks when judging the cosine distance at the hierarchical layer, without overcoming the problem of early aggregation. As a result, this may lead to an insignificant improvement in our SnoHv2. As shown in Figure 5, we find that our pruning rate on the 28-layer resgcn is higher than that of the lottery ticket, yet we can achieve relatively comparable performance. This corroborates the possibility that low-level aggregation may indeed no longer contribute to the model, allowing for early stopping at shallower layers.

## H   FUTURE WORK

**Future work.** (1) *Introducing the block concept from CV.* A relatively simple and faster way to accelerate training is to introduce the block concept from CV, combining multiple layers of adjacency matrices into one block. Within the same block, the pruning elements of all adjacency matrices are the same, and the shallow blocks align the reduced edges to the deeper layers. (2) *Designating improved early stopping strategies*, in this paper we have utilized the simplest pruning strategy to determine whether a node should stop early. We anticipate that in the future, more adaptive early stopping strategies can be discovered to assist in better supporting the Snowflake Hypothesis.

