# OpenReview forum: "The Snowflake Hypothesis: Training Deep GNN with One Node One Receptive field"
_ICLR.cc/2024/Conference — Submitted to ICLR 2024_

### Official Review · Reviewer_gnhU · 2023-10-31

**Soundness:** 2 fair
**Presentation:** 3 good
**Contribution:** 3 good
**Rating:** 5
**Confidence:** 4

**Summary:**

The manuscript "The Snowflake Hypothesis: Training Deep GNN with One Node One Receptive Field"
introduces an innovative approach, termed the Snowflake Hypothesis (SnoH), for training deep Graph
Neural Networks. This hypothesis advocates for the uniqueness of the receptive field for each node in the
graph. The authors propose a method that utilizes gradient and node-level cosine distance as metrics to
regulate the aggregation depth for each node. This approach, which can be likened to an "early stopping"
mechanism at the node level through edge pruning, has shown effectiveness across a variety of backbone
networks, offering enhanced generalizability and interpretability.

**Strengths:**

1. The paper is well-writtened, ensuring ease of comprehension. It offers a fresh perspective on tackling
the over-smoothing issue in GNN training through the early stopping method under the proposed
Snowflake Hypothesis.
2. The proposed method is not only simple and effective but also exhibits compatibility with diverse
backbone networks and training strategies. This adaptability encourages broader adoption and
potential future extensions.
3. The authors have conducted extensive experiments, encompassing different training schemes, a
variety of GNN backbones on diverse scale benchmarks. The results demonstrate that the framework
can serve as a universal operator for a range of tasks

**Weaknesses:**

1. The methodology is heavily reliant on the adjacency matrix, which confines its applicability to
message-passing based backbone networks, whose discriminative ability is limited by the 1-Weisfeiler-
Lehman Isomorphism Test. The manuscript does not offer a viable method for extending its
application to subgraph-based and multihop networks.
2. Despite claims of broad applicability, the results presented in Table 1 primarily focus on deep GNN
models, while regular network like GCN sexhibiting notable performance decline as the number of
layers increases. This observation brings into question the framework's generalizability across
different backbone networks.
3. The manuscript lacks an in-depth analysis of the impact of the parameter on the training process of
various models. Furthermore, it does not provide guidelines for setting in accordance with the
specific characteristics like the of the model and dataset.

**Questions:**

1. The paper choose cosine distance as a metric to determine when the node updation should stop. Why
was this method chosen, given its potential introduction of homogeneity assumptions, and could this
lead to a decline in performance in heterophily networks?

---

### Official Review · Reviewer_yhNV · 2023-11-01

**Soundness:** 2 fair
**Presentation:** 2 fair
**Contribution:** 2 fair
**Rating:** 3
**Confidence:** 3

**Summary:**

This paper propose SnoH, which use different receptive fields for node aggregation operation in GNN.

However, I am unconvinced that the proposed method offers significant advantages for graph learning tasks. While this method aims to boost deepGNN's performance, the enhancement seems marignal and is considerably distant from the state of the art. Additionally, the SnoH's performance on Ogbn-Arxiv appears random, which contradicts the assertion in the abstract that it "serves as a universal operator for a range of tasks and shows considerable potential with deep GNNs."

**Strengths:**

1. marginally improve deepGNN's performance.
2. evaluate the model on a range of dataset.

**Weaknesses:**

1. The motivation is unclear.
2. overclaim, SnoH does not always improve deepGNN's performance.
3. The GNN backbone is far from STOA, both on Ogbn-Proteins and Ogbn-arxiv dataset.

**Questions:**

1. The motivation is unclear, why enhance deepGNN? many GNN use 2 or 3 layers but still achieve good performance. For example, [1] use 3 layers and achieve 73\% on ogbn-arxiv, higher than the model's performance.

2. In Table 4, performance drops after using SnoHv1, so SnoHv1 is not always helpful for deepGNN.

3. Is SnoH only limited to the GNN proposed here or it also enhance other models' performance,  I would suggest evaluating SnoH on GNN  models that have demonstrated strong performance on Ogbn-Proteins and Ogbn-arxiv.


[1] Huang, Qian, et al. "Combining label propagation and simple models out-performs graph neural networks." arXiv preprint arXiv:2010.13993 (2020).

---

> ### Author Response · Authors · 2023-11-22
>
> Thank you for your thoughtful and constructive reviews of our manuscript! Based on your questions and recommendations, we give point-by-point responses to your comments and describe the revisions we made to address them.
>
> > **Weakness 1 & Question 1**: The motivation is unclear; The motivation is unclear, why enhance deepGNN? many GNN use 2 or 3 layers but still achieve good performance. For example, [1] use 3 layers and achieve 73% on ogbn-arxiv, higher than the model's performance.
>
> Thank you for your insightful question. We would like to respectfully state that, although it is true that shallow GNNs can outperform deep GNNs on certain datasets, recent explorations in deep GNNs are positioning them as the mainstream in the field of graph learning. It is noteworthy that a considerable number of SOTA methods ranking highly on the OGBN benchmark surpass not only 6 layers [1,2,3] but, in some cases, 32 layers [4]. Hence, we believe our exploration of deep GNNs contributes meaningfully to the community.
>
> > **Weakness 2 & Question 2**: overclaim, SnoH does not always improve deepGNN's performance; in Table 4, performance drops after using SnoHv1, so SnoHv1 is not always helpful for deepGNN.
>
> Thank you for your keen observation! We acknowledge that SnoHv1, as evidenced in Table 4 with the 16-layer ResGCN+, led to a slight performance decrease (0.78$\downarrow$). We faithfully reported this based on the average results from 5 runs. However, we would like to humbly state that, in numerous other scenarios (refer to Tables 1 & 5), SnoH consistently enhances the performance of various GNN backbones. Overall, we believe it is not an overclaim to assert that SnoH can bring valuable performance improvements to all GNN backbones.
>
> > **Weakness 3 & Question 3**: The GNN backbone is far from STOA, both on Ogbn-Proteins and Ogbn-arxiv dataset; Is SnoH only limited to the GNN proposed here or it also enhance other models' performance, I would suggest evaluating SnoH on GNN models that have demonstrated strong performance on Ogbn-Proteins and Ogbn-arxiv.
>
> Thank you for the constructive feedback! Indeed, the backbones (GCN/ResGCN/Res+) we selected are not the SOTA ones for current OGB datasets. However, as stated in our paper, the snowflake hypothesis is not confined to a specific GNN backbone; it is a universal operator that can be embedded in any (deep) GNN. To substantiate our standpoint, we have supplemented experiments demonstrating the application of the snowflake hypothesis to RevGNN [4] (ranked 4th and 26th on the Proteins/Products Benchmark respectively).
>
> **Table 3.** Comparisons with SnoHv2 on Ogbn-Proteins. We report an average of 3 runs.
> | Layer | 8  | 16 | 32 | 64 |
> | --- | --- | --- | --- | --- |
> | RevGNN-80 | 84.07±0.58 | 85.31±0.72 | 85.38±0.81 | 85.74±1.03 |
> | w/ SnoHv2 | 84.66±0.81 | 86.07±0.86 | 86.86±0.69 | 86.21±0.92 |
>
> **Table 3.** Comparisons with SnoHv2 on Ogbn-Products. We report an average of 3 runs.
> | Layer | 8  | 16 | 32 | 64 |
> | --- | --- | --- | --- | --- |
> | RevGNN-80 | 80.33±0.49 | 81.17±0.52 | 81.36±0.33 | 82.16±0.71 |
> | w/ SnoHv2 | 80.79±0.60 | 82.24±0.95 | 82.85±0.89 | 84.82±1.11 |
>
>
> It can be observed that SnoHv2 has brought about a substantial performance improvement for RevGNN, with the enhancement becoming more pronounced as the number of layers increases (consistent with our observations in Tables 1 & 5).
>
> ____
> **References:**
> [1] Label Deconvolution for Node Representation Learning on Large-scale Attributed Graphs against Learning Bias, Arxiv
> [2] Simplifying Graph Convolutional Networks, ICML'19
> [3] Simple and Deep Graph Convolutional Networks, ICML'20
> [4] Training Graph Neural Networks with 1000 Layers, ICML'2021

---

> > ### Comment · Reviewer_yhNV · 2023-11-23
> > **response**
> >
> > Thank you for the response, overall the improvements are marginal and the performance on Ogbn-Products/Proteins still falls short compared to the state-of-the-art, my score remains unchanged.

---

### Official Review · Reviewer_8f4f · 2023-11-03

**Soundness:** 3 good
**Presentation:** 2 fair
**Contribution:** 3 good
**Rating:** 6
**Confidence:** 3

**Summary:**

The authors propose a new training paradigm for Graph Neural Networks (GNNs): they use a masked adjacency matrix to prune the graph structure.
They evaluate on several models and datasets

**Strengths:**

Originality: as far as I can judge, this is a novel application.
Quality: empirical validation is thorough, and the results appear convincing.
Clarity: while there were some stilistic issue (see Weaknesses), the underlying thoughts were mostly clear. I was able to follow the exposition without much trouble.
Significance: given the solid results, I think this paper is relevant for the GNN community

**Weaknesses:**

The paper is solid, but currently has a lot of stylistic issues. For example, the text contains many filler-sentences ("We have another interesting observation") and hyperlatives ("These astonishing results clearly verify the effectiveness of our algorithm.").  Please go over the text, and keep in mind that good writing should have active voice, be **clear** and **concise**. There is no need to write "we meticulously conduct a multitude of experiments to validate our hypothesis..."  when one could just write "we validate our hypothesis...". This will make the text easier to read. The goal should be to convey your findings, not to convince the reviewers how awesome you are.

I'd also suggest going over the abstract: it's the most important piece of the text, and it's probably the *only* piece of your work most people will read. The current abstract contains many awkward sentences (what does "Given that the potency of numerous CV and language models is attributable to that support reliably training very deep architectures" even mean?). If available, try to work with someone who has experience in editing text and making it more readable. I think your work is overall interesting, it deserves a good, clean packaging.

I would also strongly suggest to remove the "red numbers" in Table 1, and instead add variances: The relevant part is what the average performance over 5 runs is, plus how much variance there will be between runs. The "best result out of N runs" is not valuable information, it's random-seed-lottery.

**Questions:**

What is the running time (wall clock, FLOPS) of your method vs the baselines?

---

> ### Author Response · Authors · 2023-11-22
> **Response to Reviewer 8f4f**
>
> We express our sincere thanks for the detailed and thoughtful review of our manuscript and for the encouraging appraisal of our work. We give point-by-point responses to your comments and describe the revisions we made to address them:
>
> > **Weakness 1:the text contains many filler-sentences ("We have another interesting observation") and hyperlatives ("These astonishing results clearly verify the effectiveness of our algorithm."). Please go over the text, and keep in mind that good writing should have active voice, be clear and concise. There is no need to write "we meticulously conduct a multitude of experiments to validate our hypothesis..." when one could just write "we validate our hypothesis...". This will make the text easier to read.**
>
> Thank you very much for pointing out the shortcomings in the writing of our paper. Based on your suggestions, we have read through the paper and removed the meaningless filler-sentences and revised some hyperlatives. For example, the sentence "These astonishing results clearly verify the effectiveness of our algorithm." We modify it to "These observations confirm our algorithm's effectiveness.”.
>
> > **Weakness 2:The current abstract contains many awkward sentences (what does "Given that the potency of numerous CV and language models is attributable to that support reliably training very deep architectures" even mean?)**
>
> I appreciate your insightful suggestions regarding the shortcomings in our manuscript's abstract. We have tried our best to modify the vague sentences in it. For example the sentence "Given that the potency of numerous CV and language models is attributable to that support reliably training very deep architectures", we modify it to "The success of artificial intelligence in computer vision and natural language processing largely stems from its own ability to train deep models effectively.”.
>
> > **weakness 3:I would also strongly suggest to remove the "red numbers" in Table 1, and instead add variances: The relevant part is what the average performance over 5 runs is, plus how much variance there will be between runs. The "best result out of N runs" is not valuable information, it's random-seed-lottery.**
>
> I'm grateful for for your suggestions on the experiments of our work. And I am very sorry that this is actually a mistake in our writing, all the values in Tabel1 are the average result of 5 runs, not the best result of 5 times.
>
> > **Question 1: What is the running time (wall clock, FLOPS) of your method vs the baselines?**
>
> Thanks for your insightful question. Due to limited time, we have only presented part of the experimental results(The results are shown in Table 1), and we will try our best to add more experimental results to the revised manuscript in the future.
>
> **Table 1.** Running time of different baseline using SnoHv2 on ogbn-arxiv. For each backbone we trained 500 epochs
> | baseline | original  | +SnoHv2  |
> |----------|-----------|----------|
> | GCN      | 404.67s   | 261.44s  |
> | ResGCN   | 415.49s   | 223.53s  |
> | ResGCN+  | 415.41s   | 339.49s  |
>
> Aggregation operations through adjacency matrices are a time-consuming part of graph neural networks. Therefore, the running time of our method (whether it is inference time or training time) depends on the sparsity of the adjacency matrix after pruning. The sparser the adjacency matrix, the shorter the time spent, while the original adjacency matrix is the densest.
> After adding SnoHv2, the baseline FLOPS will also decrease a bit, because the number of edges will decrease and the corresponding calculation will decrease.

---

### Official Review · Reviewer_ngCy · 2023-11-03

**Soundness:** 3 good
**Presentation:** 2 fair
**Contribution:** 3 good
**Rating:** 5
**Confidence:** 3

**Summary:**

The paper presents the "Snowflake Hypothesis" for Graph Neural Networks (GNNs) to address overfitting and oversmoothing by assigning unique receptive fields to nodes. This hypothesis is operationalized using edge pruning (via gradient and node-level distance metrics) in order to modify the receptive field / aggregation strategy for each node. The experiments show that this can be flexibly applied with different training schemes and deep GNN backbones on large-scale benchmarks.

**Strengths:**

- Tackles important problem (mitigates oversmoothing in GNNs)
- The second version of the proposed method (SnoHv2) is scale to large-scale graphs
- Extensive empirical analysis: different training schemes, datasets, GNN backbones. Results suggest that SnoHv{1,2} improve performance. This supports the hypothesis that for a subset of nodes, early stopping is necessary to mitigate oversmoothing, especially as depth increases.

**Weaknesses:**

- Paper is not well-written, would benefit from a reorganization with a shorter and more to-the-point introduction,  preliminaries and problem setup, and the algorithm describe more formally + clearly.
- Unclear if the performance improvements in general are statistically significant, an easy way to fix this is via multiple runs
- “Although DropEdge can improve performance through implicit data augmentation, it lacks interpretability in its aggregation strategy.” I am not sure how this improves interpretability over DropEdge or pruning.
- The high-level approach in SnoHv{1,2} is largely heuristic-driven and lacks grounding

**Questions:**

None, please see weaknesses.

---

### Meta-Review · Area_Chair_EKU6 · 2023-12-18

**Metareview:**

The authors propose methods to address oversmoothing of nodes in GNN. The reviewers have highlighted the importance of the problem, the new ideas, and experimental setup. However, the response did not sufficiently convince the reviewers to raise their scores and do not address all reviewer comments. The main concerns around statistical significance of the results (even the new results with error bars shown during author response are not convincing of significant boosts) and not evaluating on SOTA models is a key shortcoming that needs to be addressed in future revisions.

**Justification For Why Not Higher Score:**

The author response did not convince any of the reviewers to raise their scores. Plus couple reviewers did not even receive a response. In area chair's opinion, the main concerns around statistical significance of the results (even the new results with error bars shown during author response are not convincing of significant boosts) and not evaluating on SOTA models is a key shortcoming.

**Justification For Why Not Lower Score:**

NA

---

### Decision · Program_Chairs · 2024-01-16

Reject